# WaterSIC: Information-Theoretically (Near) Optimal Linear Layer Quantization

Egor Lifar[1]   Semyon Savkin[2]   Or Ordentlich[3]   Yury Polyanskiy[1]

## Abstract

This paper considers the problem of converting a given dense linear layer to low precision. The tradeoff between compressed length and output discrepancy is analyzed information theoretically (IT). It is shown that a popular GPTQ algorithm may have an arbitrarily large gap to the IT limit. To alleviate this problem, a novel algorithm, termed "WaterSIC", is proposed and is shown to be within a rate gap of 0.255 bits to the IT limit, uniformly over all possible covariance matrices of input activations. The key innovation of WaterSIC's is to allocate different quantization rates to different columns (in-features) of the weight matrix, mimicking the classical IT solution known as "waterfilling". Applying WaterSIC to the Llama and Qwen family of LLMs establishes new state-of-the-art performance for all quantization rates from 1 to 4 bits. Our code is available here.

## 1. Introduction

A backbone of all LLMs is the linear layer that given a column $X$ of activations outputs

$$Y = WX$$

with $W$ being a weight matrix. The main goal of the post-training quantization (PTQ) is reduction of the number of bits used for describing $W$, i.e. replacing $W$ with $\hat{W}$ (its lower resolution approximation). While dozens if not hundreds of algorithms exists at this point, most of them rely on the elementary bulding block: *layerwise quantizer*, which finds $\hat{W}$ such that $\hat{Y} = \hat{W}X \approx Y$, where the approximation quality is measured in mean-squared error, or, more rarely, in another metric (Tseng et al., 2025; Badri & Shaji, 2023).

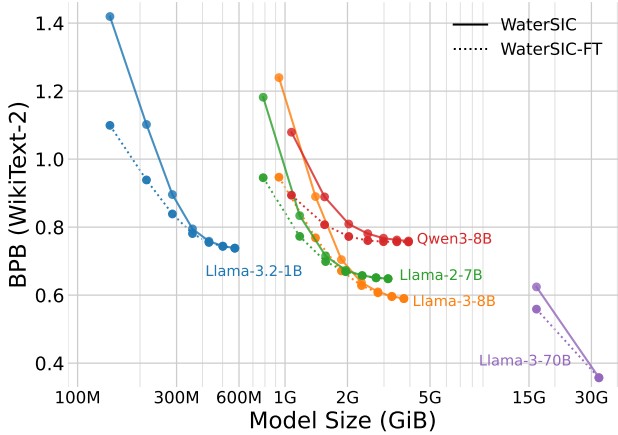

*Figure 1.* WikiText-2 bits-per-byte (BPB) vs. compressed model size (GiB) for a layerwise WaterSIC and finetuned WaterSIC-FT across multiple base models. Lines connect the same model compressed with the same algorithm and with various bit rates. Evaluated at CTX=2048.

x

Despite high-intensity research in this area, no information theoretic (IT) analysis of optimality has been attempted for the problem or the algorithms. This work proposes a new linear layer quantizer, *WaterSIC*, which is shown to achieve (on synthetic iid Gaussian weights) a gap of at most 0.25 bit to information theoretic limit. Upon applying this algorithm to real-world LLMs, we establish new SOTA in many categories. The key innovation in *WaterSIC* is using different rates for different columns (in-features) of $W$. To the best of our knowledge all existing algorithms fix the same quantization rate for all columns of $W$, in contradiction to a classical result in IT, known as *waterfilling rate allocation*.

To position our work in the research area on PTQ, let us recall that a PTQ method consists of the following ingredients:

- *Base quantizer* takes a vector of (high resolution) floating point numbers and converts it to a lower resolution version. For example, popular NVFP4 format takes 16 floats and returns 16x4 bits corresponding to individual numbers and one 8-bit microscaling coefficient.

[1]MIT [2]Independent Researcher [3]Hebrew University of Jerusalem. Correspondence to: Egor Lifar <l1far@mit.edu>.

*Proceedings of the 43rd International Conference on Machine Learning*, Seoul, South Korea. PMLR 306, 2026. Copyright 2026 by the author(s).

- *Scaling method* assigns scaling coefficient with larger granularity to make the data fit within the limits of the base quantizer. A popular strategy is *absmax*, though others are being studied (Panferov et al., 2025; Cook et al., 2025). This step is crucial for quantizing LLMs with weight outliers.

- *Calibration and GPTQ*. Instead of simply passing the whole matrix through scaling-based quantizer (a strategy known as RTN), one could adapt quantization error to statistics of activations, specifically by making it almost orthogonal to the directions of principal variation (PCA) of $X$. For that the base quantizer is applied in a clever recursive way to shape the error statistics, an idea originating from GPTQ (Frantar et al., 2023).

- *Fine-tuning*. After converting a layer to low-precision, one may run end-to-end low-precision training (Liu et al., 2025b), tune only subsequent unquantized layers (Shao et al., 2024), or adjust only some high-precision parameters of the quantized layer, e.g. scales, codebooks (Egiazarian et al., 2024).

In this work, we adopt the simplest possible setting: a uniform (round-to-nearest) base quantizer, no sophisticated scaling and no fine-tuning of any kind. Our goal is to isolate gains from our key innovation in the GPTQ step: *WaterSIC*'s unequal quantization rates. Correspondingly, the reported gains are additive with those of using more sophisticated base quantizers or adding fine-tuning. Nevertheless, even without these tweaks our evaluations (see Figs. 2 and 3) show that WaterSIC alone establishes the new SOTA.

In place of scaling, we employ entropy coding. Specifically, our base quantizer $q_\epsilon(x)$ simply rounds each coordinate independently to an equispaced $\epsilon$-grid. The resulting output consists of (possibly unbounded) integers. Instead of constraining their range via scaling, we compress this long list of integers via a high-quality data-compressor (Lempel-Ziv, Zstd or Huffman), which produces a finite (but variable) length bit-string, an idea that has been gaining popularity (Chen et al., 2026; Cheng et al., 2025; Yubeaton et al., 2025; Putzky et al., 2026), and hardware support in Blackwell (Jarmusch & Chandrasekaran, 2025).[1] When $\epsilon \to 0$ the entropy of integers grows and the length (and hence compression rate) increases. Thus by tweaking $\epsilon$ we can adjust the rate in a smooth way. In addition, entropy coding also automatically takes care of outliers: occasional large integers get assigned long bit-descriptions, but due to being infrequent do not affect the overall rate.

The key innovation of the WaterSIC is in how the scale $\epsilon$ for each channel (column) is selected. Naively, one would

take $\epsilon$ for $k$-th column to be proportional to $\sqrt{\mathrm{Var}(X_k)}$, i.e. the typical magnitude of the $k$-th activation coordinate. Surprisingly, the optimal method is to instead set $\epsilon$ to be proportional to $\sqrt{\mathrm{Var}(X_k|X_1,\ldots,X_{k-1})}$, i.e. the magnitude of independent innovation in $X_k$ not explained by linear functions of past coordinates.

The structure of the paper is as follows. In Section 3 we focus on iid Gaussian matrices $W$ and review classical IT result that shows how an overall rate budget should be allocated between different PCA directions of the covariance matrix of $X$, a so called "waterfilling" solution. Executing this strategy in its classical form is infeasible for LLMs due to constraints of the problem (as discussed below). However, another IT result (Zamir et al., 2006) demonstrates that quantizing the prediction errors with careful rate allocation can also achieve IT fundamental limit. It is then shown that in the context of LLM quantization this is equivalent to using scale choice as above, running GPTQ ane entropy coding, which is the WaterSIC algorithm. Theoretically, this shows that WaterSIC has a gap of at most 0.255 bit compared to the IT limit and does not require expensive rotations into the PCA basis. In addition, we also present a rigorous analysis of the performance of the standard GPTQ algorithm and show it can have arbitrary large gap to IT optimal limit. On the other hand, due to WaterSIC approximately implementing the waterfilling allocation, it's gap to IT limit is fully characterized by the suboptimality of the integer lattice for vector quantization of Gaussians (the reason for the 0.255 bit gap).

Section 4 describes several important tweaks of the plain WaterSIC that are critical for applying it to real-world LLMs. Section 5 describes experimental results and discusses limitations and future work. Appendices contain full details of experiments, ablations and diagnostics. We start, however, with a brief review of existing work.

## 2. Prior work on PTQ

Some PTQ works improve the base quantizer component of the algorithm. In (Elhoushi & Johnson, 2025) optimizes a scalar codebook for 4-bit quantization. (Tseng et al., 2024a) proposes an 8-dimensional vector codebook E8P of size $2^{16}$ with a fast decoding algorithm. In (Egiazarian et al., 2024), a vector codebook is used with elements obtained as sums of learned vectors. (Tseng et al., 2024b) uses trelises, which enable to use high-dimensional codebooks while maintaining fast decoding speed. (Savkin et al., 2025) proposes to use a lattice-based method, which does not require lookup tables for encoding and decoding.

Applying equivalent transformations in the network can improve the performance of quantized models. (Chee et al., 2023) rotates the model weights with Hadamard matrices,

---

[1]For example, Llama-3 family is originally in BF16 but entropy of is only about 10.5 bit/weight, mostly concentrated in the mantissa bits.

and (Ashkboos et al., 2024) uses a similar technique for activation quantization. Some methods for activation quantization ((Sun et al., 2025), (Liu et al., 2025a)) use an optimizer to find best performing transformation. (Lin et al., 2024) scales up the weights associated with important channels to reduce their quantization error, while (Xiao et al., 2023) uses scaling to balance the outliers in activations.

GPTQ (Frantar et al., 2023) yields a substantial improvement over RTN by minimizing the expected MSE of layer outputs. Prior work has explored incorporating inter-layer interactions in the objective. A number of approaches focus on Fisher information of token distributions with respect to the weight. The memory needed to store Fisher information is quadratic with respect to the weight matrix size, so it's only feasible to store its approximation. (Tseng et al., 2025) decomposes it into Kroncker product of two matrices corresponding to pairs of rows and pairs of columns, respectively. (Kim et al., 2025b) assumes that Fisher information coefficient corresponding to two parameters in different output channels is zero, and applies per-group averaging of coefficients.

Following the PTQ, one often implements fine-tuning. The most aggresive method is to continue training the quantized model over a long-phase of quantization-aware training (with optimal choice seeming to use around 10-20% of available data for that (Liu et al., 2025b)). A less expensive method is to quantize a layer, but then fine-tune subsequent unquantized layers to compensate for the quantization error (Shao et al., 2024). An even cheaper method is to quantize all layers but then run fine-tuning of only those parameters of the quantized model which are stored in floating point (scales, codebooks etc) (Tseng et al., 2024a), or at least fine-tune a few adjacent layers (Egiazarian et al., 2024).

One of the more exciting questions in the area of low-precision models is to understand Pareto frontier of the the model quality vs the total number of bits required to store weights. (Kim et al., 2025a) estimates the frontier based on weight and KV cache quantization by GPTQ, however, newer quantization algorithms achieve better model quality for the same bit budget. Other studies find often vastly different answers: ParetoQ (Liu et al., 2025b) estimates Pareto frontier at below 2 bits per parameter, whereas (Chen et al., 2026) suggests slightly above 3 bits. Memorization studies are equally inconclusive, with (Allen-Zhu & Li, 2025) estimating 2.2 bits per parameter and (Morris et al., 2025) at 3.6 bit per parameter. From information-theoretic point of view, it would be interesting to get some more robust estimates, but for that there needs to be some belief in quantization algorithm's near-optimality, which WaterSIC strongly suggests.

## 3. WaterSIC and GPTQ: Theory

Let us restrict attention to a particular linear layer in the network with weight matrix $W \in \mathbb{R}^{a \times n}$. The linear layer operates by taking (a vector of) input activations $X \in \mathbb{R}^n$ and producing $Y = WX$. The task of quantization is to replace $W$ with quantized version $\hat{W}$, which can be represented by $naR$ bits, aiming at minimizing the expected distortion

$$D = \frac{1}{na}\mathbb{E}\|(W - \hat{W})X\|_F^2$$
$$= \frac{1}{na}\mathrm{tr}(W - \hat{W})\Sigma_X(W - \hat{W})^\top, \qquad (1)$$

where the expectation is with respect to the random variable $X$ and $\Sigma_X = \mathbb{E}[XX^\top]$. In practice, the matrix $\Sigma_X$ is estimated from calibration data.

For the analysis we model the rows of $W$ as iid $\mathcal{N}(0, \sigma_W^2 I)$ random vectors in $\mathbb{R}^{1 \times n}$. While this is certainly not a precise model, it allows to derive the fundamental limits of the weight-only quantization problem, and design practical schemes that performs quite close to them. Furthermore, under this assumption, the rows of $W$ can be quantized independently without loss of optimality (assuming $n$ is large enough). We will therefore assume that $a = 1$ for the derivation of the theoretical bounds, and this is without loss of generality for $n$ large enough.

The IT limit in this case is defined as follows. Given $n > 0$, $\sigma_W^2 > 0$ and PSD matrix $\Sigma_X \in \mathbb{R}^{n \times n}$, an $(R, D, \Sigma_X)$ weight-only quantization scheme consists of an encoder $f : \mathbb{R}^n \times \mathbb{R}^{n \times n} \to \{0,1\}^*$ and decoder $g : \{0,1\}^* \to \mathbb{R}^n$, where distortion $D$ is given by (1) (with $a = 1$) and rate $R$ is defined as

$$R = \frac{1}{n}\mathbb{E}[\ell(f(W, \Sigma_X)],$$

where $\ell(\cdot)$ is the length of the binary string $(\cdot)$. The fundamental limits are defined as

$$R^*(D, \Sigma_X) = \inf\{R \,:\, \exists (R, D, \Sigma_X) - \text{scheme}\},$$
$$D^*(R, \Sigma_X) = \inf\{D \,:\, \exists (R, D, \Sigma_X) - \text{scheme}\}.$$

The main point of this section is to show that as $n \to \infty$ we have

$$D^*(R, \Sigma_X) = \sigma_W^2 |\Sigma_X|^{1/n} 2^{-2R+o(1)},$$

where $o(1)$ is with respect to $R \to \infty$ and $|\Sigma_X|$ denotes determinant of a matrix. The standard entropy-coded GPTQ (a.k.a. Huffman-GPTQ) achieves

$$D^*_{\mathrm{GPTQ}}(R, \Sigma_X) = 2^{-2R+o(1)} \frac{2\pi e}{12} \frac{\sigma_W^2}{n} \sum_{i=1}^n (\ell_{ii})^2,$$

where $\ell_{ii}$ is the $i$-th diagonal element of a lower triangular matrix of the Cholesky decomposition $\Sigma_X = LL^\top$. The

newly proposed WaterSIC, though, achieves

$$D^*_{\text{WaterSIC}}(R, \Sigma_X) = 2^{-2R+o(1)} \frac{2\pi e}{12} \sigma_W^2 \prod_{i=1}^{n} (\ell_{ii})^{\frac{2}{n}},$$

which (by AMGM) is always better than Huffman-GPTQ. However, what is more exciting is that $\prod \ell_{ii} = |L| = \sqrt{|\Sigma_X|}$ and therefore, $D^*_{\text{WaterSIC}}$ is only a factor $\frac{2\pi e}{12}$ worse than IT limit $D^*(R, \Sigma_X)$. Since $\frac{1}{2} \log_2 \frac{2\pi e}{12} = 0.255$ bit, the rate gap between WaterSIC and IT limit is at most 0.255 bit, as claimed. Another difference is that Huffman-GPTQ's performance is affected (often reduced) by applying a random rotation $U \Sigma_X U^\top$. But WaterSIC's performance is invariant to rotations (since it only depends on $|\Sigma_X|$.

### 3.1. Waterfilling lower bound

In the problem setup above, the decoder cannot depend on $\Sigma_X$. To obtain a lower bound on the smallest distortion attained by any weight-only quantizer, we consider an easier setup where $g$ can also depend on $\Sigma_X$. Any lower bound for this easier setup is clearly also a valid lower bound for our setup. Consider the singular value decomposition (SVD) $\Sigma_X = V \Lambda V^\top$. Define $\tilde{W} = WV\Lambda^{1/2}$ and $\hat{\tilde{W}} = \hat{W}(V\Lambda^{1/2})^{-1}$. With this notation, we have that

$$D = \mathbb{E} \|\tilde{W} - \hat{\tilde{W}}\|^2.$$

Thus, the problem reduces to rate $R$ quantization of a Gaussian source with independent components $\tilde{W} \sim \mathcal{N}(0, \sigma_W^2 \Lambda)$. It is well-known that the optimal rate-distortion tradeoff for this problem is given by (reverse) *waterfilling solution*[2]

$$R_{\text{WF}}(D, \Sigma_X) = \frac{1}{n} \sum_{i=1}^{n} \frac{1}{2} \log \left( \max \left\{ 1, \frac{\sigma_W^2 \lambda_i}{\tau} \right\} \right),$$

$$\text{where } D = \frac{1}{n} \sum_{i=1}^{n} \min\{\sigma_W^2 \lambda_i, \tau\}, \quad (2)$$

where $\Lambda = \text{diag}(\lambda_1, \ldots, \lambda_n)$. It is easy to see that

$$\forall D < \min\{\sigma_W^2 \lambda_i\} : R_{\text{WF}}(D, \Sigma_X) = \frac{1}{2} \log \left( \frac{\sigma_W^2 |\Sigma_X|^{\frac{1}{n}}}{D} \right)$$

$$\triangleq R_{\text{High-Rate}}(D, \Sigma_X). \quad (3)$$

To summarize, we have

**Proposition 3.1.**

$$\forall \Sigma_X \; : \; R^*(D, \Sigma_X) \geq R_{\text{WF}}(D, \Sigma_X).$$

---

[2]The standard setup is for fixed-rate quantizers, but the converse holds also for variable-rate quantizers.

We stress that the main reason it is not possible to achieve this bound in practice is because decoder needs to able to operate in the PCA basis of $\Sigma_X$, but informing decoder of the SVD basis requires sending an $n \times n$ orthogonal matrix, which is of the same order of magnitude as the original matrix $W$.

### 3.2. GPTQ and PlainWaterSIC

Applying Cholesky decomposition, we can write $\Sigma_X = LL^\top$, where $L \in \mathbb{R}^{n \times n}$ is a lower triangular matrix. Our objective function can therefore be written as

$$D = \frac{1}{n} \mathbb{E} \|Y - \hat{W}L\|^2, \; Y = WL. \quad (4)$$

Assume that the reconstruction $\hat{W}$ must belong to the scaled integer lattice $\alpha \mathbb{Z}^{1 \times n}$. Given $y \in \mathbb{R}^{1 \times n}$, the problem of finding $\hat{w} \in \alpha \mathbb{Z}^{1 \times n}$ that minimizes $\|y - \hat{w}L\|^2$ is that of finding the closest vector to $y$ in the lattice $\alpha \mathbb{Z}^{1 \times n} L$. This problem is known to be NP-hard. Thus, one must resort to sub-optimal algorithms. One such option is successive interference cancellation (SIC). This corresponds to utilizing the lower triangular structure of $L$ and sequentially deciding on the elements of $\hat{w}$ starting from $\hat{w}_n$ until reaching $\hat{w}_1$. Once we decide on the value of $\hat{w}_i$, we subtract its contribution (interference) to coordinates of $y$ that were not used for quantization yet. This procedure is described in Algorithm 1 which we refer to as the ZSIC algorithm. In ZSIC, we do not restrict ourselves to reconstruction in $\alpha \mathbb{Z}^n$, but instead allow the reconstruction to be in $\mathbb{Z}^{1 \times n} \mathcal{A}$, where $\mathcal{A}$ is a diagonal matrix with $|\mathcal{A}|^{1/n} = \alpha$. This lattice is a cartesian product $\prod_{i=1}^{n} (\alpha_i \mathbb{Z})$, whose point density is $\alpha^{-n}$ just like the lattice $\alpha \mathbb{Z}^{1 \times n}$. However, a judicious choice of $\mathcal{A}$ optimizes the spacings as a function of $L$. The benefits of this optimization will immediately become clear. Note further that Algorithm 1 is described for a matrix input, rather than a row. However, the algorithm simply applies the same quantization procedure to each one of the $a$ rows of $Y = WL$.

Denote $\text{CUBE} = \left[ -\frac{1}{2}, \frac{1}{2} \right)^{1 \times n}$. The following lemma, proved in Appendix A, characterizes the output of the ZSIC algorithm.

**Lemma 3.2.** *Assume we apply the ZSIC Algorithm 1 with* $y \in \mathbb{R}^{1 \times n}$, *lower triangular* $L \in \mathbb{R}^{n \times n}$, *and diagonal matrix* $\mathcal{A} = \text{diag}(\alpha_1, \ldots, \alpha_n) \in \mathbb{R}_+^n$. *Then*

$$e_{\text{SIC}} = y - z_{\text{SIC}} \cdot \mathcal{A}L \in \text{CUBE} \cdot \mathcal{A} \text{diag}(L).$$

Recall that in our setup the input to the ZSIC algorithm is $Y = WL$, where $W \sim \mathcal{N}(0, \sigma_W^2)$. It therefore follows that $Z_{\text{SIC}}$ is a random vector supported on $\mathbb{Z}^{1 \times n}$ and $e_{\text{SIC}}$ is a random vector supported on $\text{CUBE} \cdot \mathcal{A} \text{diag}(L)$. As

---

**Algorithm 1** ZSIC

---

**Inputs:** $Y \in \mathbb{R}^{a \times n}$, lower triangular matrix $L \in \mathbb{R}^{n \times n}$ and diagonal spacing matrix $\mathcal{A} = \mathrm{diag}(\alpha_1, \ldots, \alpha_n) \in \mathbb{R}_+^{n \times n}$.
**Outputs:** $Z_{\mathrm{SIC}} \in \mathbb{Z}^{a \times n}$      $\triangleright Z_{\mathrm{SIC}}\mathcal{A} \approx \mathrm{argmin}_Z \|Y - Z\mathcal{A}L\|_2^2$

$Z_{\mathrm{SIC}} \leftarrow 0^{a \times n}$      $\triangleright$ Initialize $Z_{\mathrm{SIC}}$ with zeros
**for** $i = n : 1$ **do**
    $Z_{\mathrm{SIC},:,i} \leftarrow \mathrm{round}\left(\frac{Y_{:,i}}{\alpha_i \ell_{ii}}\right)$      $\triangleright$ $Y_{:,i}$ and $Z_{\mathrm{SIC},:,i}$ is the $i$th column of $Y$ and $Z_{\mathrm{SIC}}$, respectively
    $Y \leftarrow Y - \alpha_i Z_{\mathrm{SIC},:,i} \cdot L_{i,:}$      $\triangleright$ $L_{i,:}$ is the $i$th row of $L$
**end for**

---

the ratio $\sigma_W / \det |\mathcal{A}|$ grows, the distribution of $e_{\mathrm{SIC}}$ approaches the uniform distribution on $\mathrm{CUBE} \cdot \mathcal{A} \, \mathrm{diag}(L)$ (see Appendix B) and we obtain

$$D_{\mathrm{SIC}} \approx \frac{1}{n}\frac{1}{12}\sum_{i=1}^{n}(\alpha_i \ell_{ii})^2 \tag{5}$$

$$= \frac{|\mathcal{A}L|^{2/n}}{12} \frac{\frac{1}{n}\sum_{i=1}^{n}(\alpha_i \ell_{ii})^2}{\prod_{i=1}^{n}(\alpha_i \ell_{ii})^{2/n}} \tag{6}$$

$$= |\mathcal{A}|^{2/n}\frac{|\Sigma_X|^{1/n}}{12}\frac{\frac{1}{n}\sum_{i=1}^{n}(\alpha_i \ell_{ii})^2}{\prod_{i=1}^{n}(\alpha_i \ell_{ii})^{2/n}} \tag{7}$$

Denote by $H(\cdot)$ the Shannon entropy of a discrete random variable and by $h(\cdot)$ the differential entropy of a continuous random variable. As $\sigma_W / \det |\mathcal{A}|$ grows, we also have that

$$\sum_{i=1}^{n}H(Z_{\mathrm{SIC},i}) \approx nh(W) - \log|\mathcal{A}| \tag{8}$$

$$= \frac{n}{2}\log\left(\frac{2\pi e \sigma_W^2}{|\mathcal{A}|^{2/n}}\right) \tag{9}$$

$$\approx \frac{n}{2}\log\left(\frac{|\Sigma_X|^{1/n}\sigma_W^2}{D_{\mathrm{SIC}}}\frac{2\pi e}{12}\frac{\frac{1}{n}\sum_{i=1}^{n}(\alpha_i \ell_{ii})^2}{\prod_{i=1}^{n}(\alpha_i \ell_{ii})^{2/n}}\right), \tag{10}$$

where (8) is derived in Appendix B, and in (10) we have used (7).

If we apply the ZSIC algorithm on a matrix $Y = WL$, where the rows of $W$ are independently drawn from $\mathcal{N}(0, \sigma_W^2 I)$ (more discussion on weight Guassianity is in Appendix E), each column $Z_{\mathrm{SIC},:,i}$ of the algorithm's output will consist of iid entries from a distribution on $\mathbb{Z}$ with entropy $H(Z_{\mathrm{SIC},i})$. Thus, if the number of rows $a$ is sufficiently large, encoding $Z_{\mathrm{SIC},:,i}$ to bits can be done via any standard entropy coding algorithm, and the expected number of bits in this description will be $aH(Z_{\mathrm{SIC},i})(1 + o(1))$, where $o(1)$ vanishes as $a \to \infty$. From the entropy coded binary descriptions of all $n$ columns, the decoder can recover $Z_{\mathrm{SIC}}$ and set $\hat{W} = Z_{\mathrm{SIC}}\mathcal{A}$. If the matrix $\mathcal{A}$ is determined by the encoder, it also needs to describe the entries $\alpha_1, \ldots \alpha_n$ in bits. Since those are only $n$ scalars, the cost of their description is negligible for $a \gg 1$.

It therefore follows that applying ZSIC algorithm with matrix $\mathcal{A} = \mathrm{diag}(\alpha_1, \ldots, \alpha_n)$ and describing each column

of the resulting matrix $Z_{\mathrm{SIC}}$ using entropy coding approximately achieves the following tradeoff between rate and distortion

$$R_{\mathrm{SIC}}(D, \Sigma_X) \approx R_{\mathrm{High-Rate}}(D, \Sigma_X)$$
$$+ \frac{1}{2}\log\left(\frac{2\pi e}{12}\right) + \frac{1}{2}\log\left(\frac{\frac{1}{n}\sum_{i=1}^{n}(\alpha_i \ell_{ii})^2}{\prod_{i=1}^{n}(\alpha_i \ell_{ii})^{2/n}}\right), \tag{11}$$

for $D$ small enough.

The canonical GPTQ algorithm is completely equivalent to the ZSIC algorithm with $\mathcal{A} = \alpha I$ (Chen et al., 2026; Birnick, 2026). When the matrix $Z_{\mathrm{SIC}}$ is further described using entropy coding, we get the *Huffman-GPTQ* algorithm explicitly proposed in (Chen et al., 2026) under the name *HPTQ*, which we use interchangeably with Huffman-GPTQ.

The last term in (11) is non-negative due to the arithmetic mean-geometric Mean (AMGM) inequality. It therefore follows that the choice $\mathcal{A} = \alpha I$ is sub-optimal in general, and the optimal choice (under the approximations leading to (11)) is

$$\alpha_i = \frac{c}{|\ell_{ii}|}, \quad i = 1, \ldots, n, \tag{12}$$

where $c > 0$ is a constant that determines the density of the lattice $\mathbb{Z}^{1 \times n}\mathcal{A}$. In particular, if we want the density to be $\alpha^{-n}$, as for the lattice $\alpha\mathbb{Z}^{1 \times n}$, we choose $c = \alpha \cdot |L|^{1/n}$.

We refer to the resulting algorithm for weight-only quantization that utilizes ZSIC together with this choice of $\mathcal{A}$ and entropy coding as PlainWaterSIC, and it is described in Algorithm 2. In the next section we describe several improvements of this algorithm, that result in the full WaterSIC, see Algorithm 3. Note that if we modify the computation of $\alpha_i$ in Algorithm 2 to $\alpha_i = \alpha\forall i = 1, \ldots, n$, we get the HPTQ algorithm from (Chen et al., 2026).

The expression (11) for the rate-distortion tradeoff a ZSIC relied on approximations. The next theorem, proved in Appendix B, shows that these approximations become exact in the limit of high-rate/low-distortion.

---

**Algorithm 2** PlainWaterSIC weight-only quantization

---

**Inputs:** $W \in \mathbb{R}^{a \times n}$, PSD matrix $\Sigma_X$ and point density $\alpha > 0$.
**Outputs:** $(\alpha_1, \ldots, \alpha_n) \in \mathbb{R}_+^n$, and binary vectors $B_1, \ldots, B_n \in \{0, 1\}^*$ encoding the $n$ columns of $Z_{\text{SIC}}$ s.t. $\hat{W} = Z_{\text{SIC}} \text{diag}(\alpha_1, \ldots, \alpha_n)$.

Compute lower-triangular $L \in \mathbb{R}^{n \times n}$ such that $\Sigma_X = LL^\top$          ▷ Using the Cholesky decomposition
$\alpha_i \leftarrow \alpha \cdot \frac{|L|^{\frac{1}{n}}}{|\ell_{ii}|}, \; \forall i \in [n]$          ▷ $\ell_{ii}$ is the $i$th diagonal element of $L$
$\mathcal{A} \leftarrow \text{diag}(\alpha_1, \ldots, \alpha_n)$
$Z_{\text{SIC}} \leftarrow \text{ZSIC}(WL, L, \mathcal{A})$          ▷ Apply Algorithm 1 with arguments $Y = WL$, $L$ and $\mathcal{A}$
**for** $i = 1 : n$ **do**
    $B_i \leftarrow \text{EC}(Z_{\text{SIC},:,i})$          ▷ Entropy coding for the $i$th column of $Z_{\text{SIC},:,i}$
**end for**

---

**Theorem 3.3.** *For any positive semi definite matrix $\Sigma_X$*

$$\lim_{a \to \infty} \lim_{D \to 0} R_{\text{GPTQ}}(D, \Sigma_X) - R_{\text{WF}}(D, \Sigma_X)$$
$$= \frac{1}{2} \log \left( \frac{2\pi e}{12} \right) + \frac{1}{2} \log \left( \frac{\frac{1}{n} \sum_{i=1}^n (\ell_{ii})^2}{\prod_{i=1}^n (\ell_{ii})^{2/n}} \right) \quad (13)$$

*and*

$$\lim_{a \to \infty} \lim_{D \to 0} R_{\text{WaterSIC}}(D, \Sigma_X) - R_{\text{WF}}(D, \Sigma_X)$$
$$= \frac{1}{2} \log \left( \frac{2\pi e}{12} \right) \quad (14)$$

## 4. WaterSIC: full algorithm

Now, in previous section a conceptual version of our algorithm, PlainWaterSIC, was described. Using it as motivation, we now describe the actual algorithm used for compressing linear layers. As before, we will have $\Sigma_X = LL^T$, where $L$ is lower-triangular. Given a constant $c$ we set $i$-th column grid spacing to $\alpha_i = \frac{c}{L_{i,i}}$. We now list series of modifications we will make.

**LMMSE correction**. When ZSIC (Algorithm 1) calls a round($\cdot$) on a $i$-th column of the matrix $Y = WL$, it effectively solves

$$\underset{z \in \mathbb{Z}^n}{\text{argmin}} \sum_{k=1}^a (Y_{k,i} - z_k L_{i,i} \alpha_i)^2 = \text{round}(Y_{:,i}/c)$$

and results in the approximation $Y_{:,i} \approx zc$. However, since entries of $Y_{:,i}$ are typically unimodal with a mode at 0, the rounding errors tend to be biased away from 0. Thus, a better reconstruction of $Y_{:,i}/c$ can be obtained by applying an extra shrinkage factor $\gamma_i$ replacing $zc$ with $\gamma_i zc$, where scalar $\gamma_i$ (known as linear-MMSE, or LMMSE correction) can be chosen via solving a simple quadratic equation:

$$\gamma_i = \underset{\gamma \in \mathbb{R}}{\text{argmin}} \sum_{k=1}^a (Y_{k,i} - \gamma c z_k)^2 = \frac{\sum_k Y_{k,i} z_k}{c \sum_k z_k^2}. \quad (15)$$

Importantly, recursive ZSIC adjustment to $Y$ should use the $\gamma_i$-corrected value too: $Y \leftarrow Y - \gamma_i c L_{i,:} z$.

Note that overall we see that the final weight $\hat{w}_i \in (\alpha_i \gamma_i) \mathbb{Z}^n$ and since $\gamma_i < 1$ one may wonder why not use the finer grid $(\alpha_i \gamma_i)$ from the start? The answer is that reducing the grid spacing will increase the entropy of $\hat{w}_i$ (and rate). This is one of the counter-intuitive effects in the low-rate regime: it is unnatural, but important, to dequantize to elements that are distinct from the quantization ones (unless one uses a sophisticated vector quantizer, i.e. a centroidal Voronoi tesselation (CVT), such as returned by a fully converged Lloyd-Max algorithm).

**Activation drift correction (Qronos/QA-LDLQ).** Several works (Savkin et al., 2025) (Section 4.5), (Zhang et al., 2026), (Zhang et al., 2025) simultaneously noticed the following discrepancy in the original formulation: instead of minimizing $\mathbb{E}[\|WX - \hat{W}X\|_2^2]$ we should be minimizing

$$\min_{\hat{W}} \mathbb{E}[\|WX - \hat{W}\hat{X}\|_2^2],$$

where $\hat{X}$ is the input activation in the quantized model (i.e. due to quantization of previous layers, the input $\hat{X}$ to the present layer is different from $X$ in the unquantized model). This problem after some algebra reduces to

$$\min_{\hat{W}} \|\hat{y} - \hat{W}\hat{L}\|_2^2, \quad (16)$$

where $\Sigma_{\hat{X}} = \mathbb{E}[\hat{X}\hat{X}^\top]$ and its Cholesky decomposition is $\Sigma_{\hat{X}} = \hat{L}\hat{L}^\top$, $\Sigma_{X,\hat{X}} = \mathbb{E}[X\hat{X}^\top]$ and

$$\hat{y} = W\Sigma_{X\hat{X}}(\hat{L}^\top)^{-1}. \quad (17)$$

Thus, the only modification to the previous discussion is the replacement of the Hessian and adjusted value of $y$.

**Residual stream correction.** We noticed that some of the hardest to compress layers are the downprojection layers (wo for attention and w2 for FFN). Part of the reason is that they in fact do not produce output $Y = WX$ as we have

been assuming, but rather $Y = WX + R$, where $R$ is the state of the residual stream to which the downprojection contributes to. For this reason, we should really be solving the problem

$$\min_{\hat{W}} \mathbb{E}[\|WX + R - (\hat{W}\hat{X} + \hat{R})\|_2^2],$$

where $\hat{X}, \hat{R}$ are the input activations and the state of residual stream in the quantized model. In this case, after introducing the $\Sigma_{\Delta,\hat{X}} = \mathbb{E}[(R - \hat{R})\hat{X}^\top]$ we can see that the problem reduces to (16) but with

$$\hat{y} = (W\Sigma_{X,\hat{X}} + \Sigma_{\Delta,\hat{X}})(\hat{L}^\top)^{-1}. \quad (18)$$

**Diagonal rescalers.** The final optimization we would like to do is another round of row/column rescalers. Specifically, after running the ZSIC algorithm, we obtain a diagonal matrix $\mathcal{A}$ and an integer matrix $Z_{\text{SIC}}$. We set $\hat{W}_0 = Z_{\text{SIC}}\mathcal{A}$ as our preliminary estimate and search the final optimizer in the form

$$\hat{W} = T\hat{W}_0\Gamma,$$

where $T = \text{diag}(t_1, \ldots, t_a)$ and $\Gamma = \text{diag}(\gamma_1, \ldots, \gamma_n)$. Because of scale invariance, we will also normalize the matrices such that $\text{tr}\, T = a$. Initially, we set $T = I$ and $\Gamma$ is taken from values $\gamma_i$ obtained in (15).

The loss objective to be minimized in terms of $T$ and $\Gamma$ is

$$\ell(T, \Gamma) = \text{tr}(W\Sigma_X W^\top - 2(W\Sigma_{X,\hat{X}} + \Sigma_{\Delta,\hat{X}})\Gamma\hat{W}_0^\top T).$$

For a fixed $\Gamma$ the loss in terms of $T = \text{diag}(t_1, \ldots, t_a)$ is given by

$$\sum_{i=1}^{a} t_i^2 F_{i,i}^{(8)} - 2F_{i,i}^{(7t)} t_i,$$

where $F^{(8)} = \hat{W}_0\Gamma\Sigma_{\hat{X}}\Gamma\hat{W}_0^\top$, $F^{(7t)} = (W\Sigma_{X,\hat{X}} + \Sigma_{\Delta,\hat{X}})\Gamma\hat{W}_0^\top$. Thus, the optimal value of $t_i$ can be given by $t_i = F_{i,i}^{(7t)}/F_{i,i}^{(8)}$.

On the other hand, for a fixed $T$ the loss function in terms of $\Gamma = \text{diag}(\gamma_1, \ldots, \gamma_n)$ is given by

$$\sum_{i,j} \gamma_i \gamma_j F_{i,j}^{(3)} - 2\sum_i F_{i,i}^{(4)} \gamma_i,$$

where $F^{(3)} = F^{(2)} \odot \Sigma_{\hat{X}}$, $F^{(2)} = \hat{W}_0^\top T^2 \hat{W}_0$, $F^{(4)} = \hat{W}_0^\top T(W\Sigma_{X,\hat{X}} + \Sigma_{\Delta,\hat{X}})$. Note that by Schur's product theorem $F^{(3)}$ is positive definite whenever $F^{(2)}$ and $\Sigma_{\hat{X}}$ are. Thus, minimizer of this expression exists and is given by

$$(\gamma_1, \ldots, \gamma_n)^\top = F^{(3)-1} \text{diag}(F^{(4)}).$$

Thus by alternating $T$ and $\Gamma$ steps, we can progressively improve the loss attained by the $\hat{W}$. (We also need to renormalize $T$ and $\Gamma$ after each step so that $\text{tr}\, T = a$ is satisfied.)

Note also that adding a row rescaler (in BF16) adds $16/a$ overhead to the final rate (and recall that $16/n$ is paid for communicating $\mathcal{A}$, which we fuse into $\mathcal{A}\Gamma$).

See Fig. 4 for representation of typical values of $T$ and $\Gamma$.

**Entropy coding.** In Algorithm 2 we performed entropy coding on each column of $Z_{\text{SIC}}$ separately. This makes sense because for iid rows of $W$, each column of $Z_{\text{SIC}}$ is iid, but the distribution may be different from column to column. In practice, however, we observed that if instead of this we perform entropy coding jointly on the entire matrix $Z_{\text{SIC}}$, the increase in rate is negligible. Thus, in the practical WaterSIC algo we go with the latter option. Note that this does not imply that most columns are coded to the same rate, which is indeed not the case (see Fig. 5).

We also checked that instead of entropy coding, one might use LZMA or Zstd compressor and indeed obtain the same number of bits as predicted by our estimate of entropy over entries. We report these findings in the Appendix, E.

**Rate assignment.** To achieve a desired target rate for the layer, we note that the final entropy is a monotone function of the constant $c$, which is approximately linear with a slope close to unity. We exploit this structure via a secant method: evaluating at the target rate reveals an offset, which a secant correction refines in about 3 iterations to within $< 0.005$ bits of the target. For computational efficiency, the entropy estimate uses a randomly sampled fraction of the rows. We also maintain a global budget that can be allocated to the remaining unquantized layers, which mitigates discrepancies between the estimated and actual entropy.

**Attention-weighted calibration.** The covariance matrices $\Sigma_X$, $\Sigma_{\hat{X}}$, and $\Sigma_{X,\hat{X}}$ above are estimated by averaging uniformly over all token positions. When quantizing attention matrices $W_K, W_Q, W_V$, however, we found out that some tokens need to be quantized with higher fidelity. To correct for this, we weight the covariance estimates for QKV projections by a per-token attention importance score:

$$p_j := \frac{1}{N_H(T-j)}\sum_{h=1}^{N_H}\sum_{i=j}^{T-1} \alpha_{h,i,j}. \quad (19)$$

Here, $\alpha_{h,i,j}$ is the attention probability from query $i$ to key $j$ in head $h$, computed from the unquantized model. The weighted covariances $\Sigma_X^{(w)} = \mathbb{E}\left[\sum_j p_j x_j x_j^\top\right]$ (and analogously $\Sigma_{\hat{X}}^{(w)}$ and $\Sigma_{X,\hat{X}}^{(w)}$) are then substituted into (16)–(17). This weighting applies only to QKV projections.

**Adaptive mixing.** We noticed that when accumulated prior layers introduce anomalously large discrepancy between quantized $\hat{X}$ and unquantized $X$ inputs, the drift compensation and attention weighting become overly fixated on

correcting this discrepancy, thus losing fidelity on truly relevant features of $X$. To fix this we can replace $\hat{X}$ with $X$ with a certain probability $\epsilon_{\text{qr}}$ during calibration calculation. Similarly, we introduce parameter $\epsilon_{\text{aw}}$ that determines the amout of attention-weighting. Overall, the actual calibration matrices we use are given by

$$
\begin{aligned}
\Sigma_X^{(\text{final})} &:= (1 - \epsilon_{\text{aw}}) \, \Sigma_X^{(w)} + \epsilon_{\text{aw}} \Sigma_X \,, \\
\Sigma_\bullet^{(\text{final})} &:= (1 - \epsilon_{\text{aw}}) \left( (1 - \epsilon_{\text{qr}}) \Sigma_\bullet^{(w)} + \epsilon_{\text{qr}} \Sigma_X^{(w)} \right) + \\
&\quad \epsilon_{\text{aw}} ((1 - \epsilon_{\text{qr}}) \Sigma_\bullet + \epsilon_{\text{qr}} \Sigma_X) \,, \tag{20}
\end{aligned}
$$

where $\bullet$ in the last equation stands for $\hat{X}$ or $X, \hat{X}$ (that is, gives equation for effective $\Sigma_{\hat{X}}$ and $\Sigma_{X, \hat{X}}$). Parameters $\epsilon_{\text{qr}}$ and $\epsilon_{\text{aw}}$ are not set apriori but are optimized using golden-search algorithm for each layer, see details in Appendix C and Tables 3, 4.

We only applied mixing optimizations for attention projections $(w_q, w_k, w_v)$, for the rest $\epsilon_{\text{qr}} = 0$ and $\epsilon_{\text{aw}} = 1$.

**Dead feature erasure.** We found that occasionally some input dimensions of $X$ have near-zero variance, i.e., $[\Sigma_X]_{ii} \approx 0$. These "dead features" cause numerical issues in the Cholesky decomposition of $\Sigma_{\hat{X}}$ and can lead to unstable rescaler optimization, while carrying negligible signal. Traditionally, these were handled by *damping*, which adds a multiple of identity to certain matrices (see Appendix C), but we apply a more direct workaround. We declare dimension $i$ dead if $[\Sigma_X]_{ii} < \tau \, \tilde{\sigma}^2$, where $\tilde{\sigma}^2 := \text{median}_j [\Sigma_X]_{jj}$ and $\tau = 10^{-3}$. We use the median rather than the mean because a few high-variance dimensions (e.g., in SiLU-gated intermediate activations) can inflate the mean by orders of magnitude, causing the threshold to erroneously flag the majority of dimensions as dead. We then set the corresponding columns of $W$ to zero and apply our gquantization pipeline to the reduced system obtained by removing these dimensions. The quantized weight is expanded back to the original size by inserting zero-columns at the dead features. In early layers, the number of dead dimensions can be substantial (e.g., due to layer normalization suppressing certain coordinates), which both improves the numerical stability of the Cholesky factor $\hat{L}$ and frees quantization rate budget for the remaining live dimensions.

**Post-quantization finetuning.** After the quantization pipeline produces integer codes $Z$ and initial rescalers $T, \Gamma$, we can further improve quality by finetuning the continuous parameters $t_1, \ldots, t_a$ and $\gamma_1, \ldots, \gamma_n$ via gradient descent, while keeping the integer codes $Z$ frozen. Since the dequantized weight $\hat{W} = T(Z \odot \alpha) \Gamma$ is fully differentiable with respect to $t$ and $\gamma$, no straight-through estimator is needed. We use distillation loss, i.e. minimize the KL divergence between the unquantized (teacher) and quantized (student) model outputs:

$$
\mathcal{L} = \text{KL}(p_{\text{teacher}}(\cdot \mid x) \| p_{\text{student}}(\cdot \mid x))
$$

using AdamW with cosine annealing. The total number of trainable parameters is $a + n$ per layer (two vectors), which is negligible compared to the $an$ frozen integer codes. As shown in Table 2, training these additional parameters under the cross-entropy loss even allowed us to improve upon the base model's perplexity.

In the plots and tables, we make a distinction between Water-SIC and WaterSIC-FT versions of the algorithm, where the latter one uses post-quantization finetuning and the former one does not.

The full algorithm is detailed in the appendix; see Algorithms 3 and 4. In Appendix E, we additionally analyze how the individual techniques affect quantization quality by reporting relative MSE plots for the inputs to each layer's weight matrices.

## 5. Evaluation results and limitations

To evaluate the performance of WaterSIC, we focus on two representative dense LLMs: Llama-3.2-1B and Qwen3-8B. Throughout, we report WikiText-2 validation perplexity at context length 2048 as a function of the average bitrate. In our plots, WaterSIC and other entropy-coded methods report rates in bits/weight via entropy, whereas several prior baselines report rates via log-cardinality; we follow the conventions used in the respective references. In Figure 1, we compare the performance of WaterSIC across various models and bitrates.

**Llama-3.2-1B.** Figure 2 and Table 1 shows the perplexity-rate tradeoff for Llama-3.2-1B. We compare WaterSIC to a diverse set of PTQ baselines: AWQ (Lin et al., 2024), Huffman-GPTQ (Chen et al., 2026), NestQuant (Savkin et al., 2025), and QTIP (Tseng et al., 2024b). Across the range of rates we consider, WaterSIC improves upon the best available baselines at comparable bitrate, yielding a consistently better frontier in the low-to-mid rate regime. A comparison between WaterSIC and Huffman-GPTQ on suite of zero-shot accuracy benchmarks on Llama-3.2-1B across multiple rates is given in Table 17 in Appendix G.

**Qwen3-8B.** Figure 3 and Table 2 summarize results on Qwen3-8B. We compare against classical weight-only baselines such as GPTQ (Frantar et al., 2023) and RTN, and against entropy-based baselines (Huffman-GPTQ, Huffman-RTN) as reported and evaluated by Chen et al. (2026). WaterSIC attains state-of-the-art perplexity at all rates, demonstrating that our method performs well on a model outside of Llama family. A comparison between WaterSIC and Huffman-GPTQ, as well as other schemes, on suite of zero-shot accuracy benchmarks on Llama-3.2-1B across multiple rates is given in Table 18 in Appendix G.

In Appendix F, we further report additional quantization

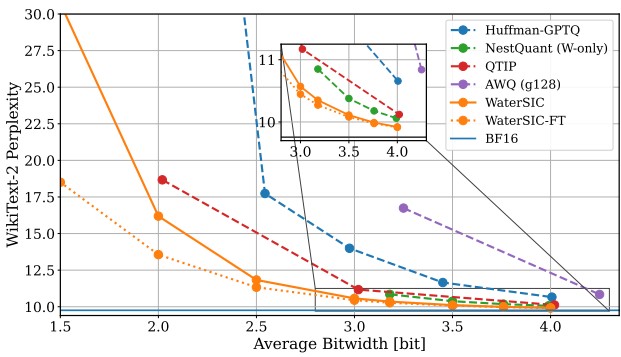

*Figure 2.* Llama-3.2-1B: WaterSIC and WaterSIC-FT vs other algorithms. WaterSIC, WaterSIC-FT and Huffman-GPTQ use entropy to report rates, others use log-cardinality. Evaluated at CTX=2048.

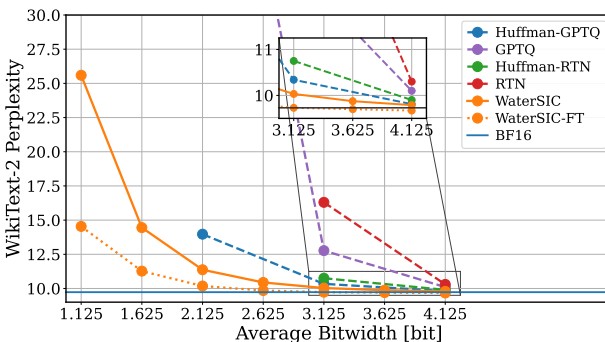

*Figure 3.* Qwen3-8B: WaterSIC and WaterSIC-FT vs other algorithms. WaterSIC, WaterSIC-FT, Huffman-GPTQ and Huffman-RTN use entropy to report rates, others use log-cardinality. Evaluated at CTX=2048.

*Table 1.* Comparison of wikitext2 perplexity results on Llama3.2-1B. In each group of rows WaterSIC-FT achieves the best PPL while having minimal rate. The unquantized model perplexity is 9.76. Evaluated at CTX=2048.

| Method | Avg. Bitwidth | WikiText-2 PPL ↓ |
|---|---|---|
| WaterSIC | 1.00 | 82.44 |
| WaterSIC-FT | **1.00** | **30.47** |
| WaterSIC-FT | **1.50** | **18.50** |
| WaterSIC | 1.50 | 30.73 |
| Huffman-GPTQ | 1.52 | 1000+ |
| Huffman-GPTQ | 1.94 | 86.80 |
| WaterSIC-FT | **2.00** | **13.59** |
| WaterSIC | 2.00 | 16.19 |
| QTIP | 2.02 | 18.67 |
| WaterSIC-FT | **2.50** | **11.33** |
| WaterSIC | 2.50 | 11.83 |
| Huffman-GPTQ | 2.54 | 17.74 |
| Huffman-GPTQ | 2.97 | 13.99 |
| WaterSIC-FT | **3.00** | **10.45** |
| WaterSIC | 3.00 | 10.57 |
| QTIP | 3.02 | 11.17 |
| NestQuant (W-only) | 3.18 | 10.85 |
| WaterSIC-FT | **3.18** | **10.27** |
| WaterSIC | 3.18 | 10.35 |
| AWQ (g128) | 3.25 | 16.74 |
| Huffman-GPTQ | 3.45 | 11.65 |
| WaterSIC-FT | **3.50** | **10.08** |
| WaterSIC | 3.50 | 10.11 |
| NestQuant (W-only) | 3.50 | 10.38 |
| NestQuant (W-only) | 3.76 | 10.18 |
| WaterSIC-FT | **3.76** | **9.98** |
| WaterSIC | 3.76 | 9.99 |
| NestQuant (W-only) | 3.99 | 10.06 |
| WaterSIC-FT | **4.00** | **9.91** |
| WaterSIC | 4.00 | 9.92 |
| Huffman-GPTQ | 4.01 | 10.66 |
| QTIP | 4.02 | 10.12 |
| AWQ (g128) | 4.25 | 10.84 |

| Method (bits) | 2.125 | 2.625 | 3.125 | 3.625 | 4.125 |
|---|---|---|---|---|---|
| Huffman-GPTQ | 13.97 | – | 10.34 | – | 9.81 |
| GPTQ | 57.51 | – | 12.77 | – | 10.10 |
| Huffman-RTN | 593.05 | – | 10.75 | – | 9.90 |
| RTN | $2 \times 10^{10}$ | – | 16.30 | – | 10.30 |
| WaterSIC | 11.37 | 10.44 | 10.03 | 9.87 | 9.79 |
| WaterSIC-FT | **10.18** | **9.85** | **9.73** | **9.70** | **9.67** |

*Table 2.* Qwen3-8B WikiText-2 perplexity (lower is better) at fixed average bitwidths. Results for Huffman-GPTQ(HPTQ)/GPTQ/Huffman-RTN(HRTN)/RTN are taken from Chen et al. (2026); WaterSIC-FT achieves the best perplexity at all rates. The unquantized model perplexity is 9.73. Evaluated at CTX=2048.

Llama-2-7B (Table 11 and Figure 14) models.

**Limitations and future work.** Our evaluation focuses on post-training weight quantization and does not include several complementary directions. First, we only focused on layerwise optimization of Euclidean post-matmul loss. Some of the best modern algorithms, however, attempt to better approximate target PPL (or KL) loss, cf. YAQA (Tseng et al., 2025). It would be quite interesting to incorporate this idea into WaterSIC. Second, while we rely on entropy estimates and standard lossless codecs to translate entropy rates into compressed bitstreams, we did not benchmark end-to-end (de)compression throughput and its interaction with real hardware kernels. Third, current experiments are conducted on relatively small to mid-sized models, and scaling to substantially larger models may expose new practical bottlenecks.

results for the Llama-3-8B (Table 9 and Figure 13) and

## Impact Statement

This paper presents work whose goal is to advance the field of Machine Learning. There are many potential societal consequences of our work, none which we feel must be specifically highlighted here.

## Acknowledgement

This material is based upon work supported by the National Science Foundation under Grant No CCF-21-12665. The work of EL and YP was supported (in part) by the generous gifts from Jane Street and Google Research. The work of OO was supported by the Israel Science Foundation (ISF), grant No. 2878/25.

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

## A. Proof of Lemma 3.2

Let $z_{\text{SIC}}(y, L, \mathcal{A})$ be the result of applying the SIC algorithm with inputs $y \in \mathbb{R}^{1 \times n}$ and $L, \mathcal{A} \in \mathbb{R}^{n \times n}$. The lemma follows from combining the two following observations:

1. The region of all $y \in \mathbb{R}^{1 \times n}$ that are mapped to $0$ is

$$\left\{ y \in \mathbb{R}^{1 \times n} : z_{\text{SIC}}(y, L, \mathcal{A}) = 0 \right\} = \text{CUBE} \cdot \mathcal{A} \operatorname{diag}(L)$$

2. For any $z \in \mathbb{Z}^{1 \times n}$ it holds that

$$z_{\text{SIC}}(y + z\mathcal{A}L, L, \mathcal{A}) = z\mathcal{A} + z_{\text{SIC}}(y, L, \mathcal{A}).$$

Thus, the ZSIC algorithm induces the partition of space to decision regions

$$\mathcal{D}_z = z\mathcal{A}L + \text{CUBE} \cdot \mathcal{A} \operatorname{diag}(L), \ \forall z \in \mathbb{Z}^{1 \times n} \tag{21}$$

and $z_{\text{SIC}}(y, L, \mathcal{A}) = z\mathcal{A}$ iff $y \in \mathcal{D}_z$. Consequently, $e_{\text{SIC}} = y - z_{\text{SIC}}(y, L, \mathcal{A}) \in \text{CUBE} \cdot \mathcal{A} \operatorname{diag}(L)$.

## B. Proof of Theorem 3.3

### B.1. Preliminaries

**Flatness factor of a lattice:** For a lattice $\Lambda \subset \mathbb{R}^{1 \times n}$ and $\sigma > 0$, define the $\Lambda$-periodic function $f_{\sigma, \Lambda} : \mathbb{R}^{1 \times n} \to \mathbb{R}$

$$f_{\sigma, \Lambda}(x) = \sum_{\lambda \in \Lambda} \phi_\sigma(x - \lambda), \tag{22}$$

where

$$\phi_\sigma(x) = (2\pi\sigma^2)^{-n/2} e^{-\frac{\|x\|^2}{2\sigma^2}} \tag{23}$$

The flatness factor of a lattice $\Lambda$ is defined as (Micciancio & Regev, 2007; Ling et al., 2014)

$$\epsilon_\Lambda(\sigma) = \sup_{x \in \mathbb{R}^{1 \times n}} \left| \frac{f_{\sigma, \Lambda}(x)}{1/\operatorname{covol}(\Lambda)} - 1 \right|. \tag{24}$$

It is well-known, see e.g. (Ling et al., 2014, Corollary 1) that

$$\epsilon_\Lambda(\sigma) = \sum_{\lambda' \in \Lambda^* \setminus \{0\}} e^{-2\pi^2 \sigma^2 \|\lambda'\|^2}, \tag{25}$$

where $\Lambda^*$ is the dual lattice of $\Lambda$. Note that the dual lattice of $\mathbb{Z}^{1 \times n}$ is $\mathbb{Z}^{1 \times n}$. Furthermore, if $\tilde{\mathcal{A}} = \operatorname{diag}(\tilde{\alpha}_1, \ldots, \tilde{\alpha}_n)$ for fixed $\tilde{\alpha}_1, \ldots, \tilde{\alpha}_n > 0$, and $\tilde{\Lambda} = \alpha \mathbb{Z}^{1 \times n} \tilde{\mathcal{A}}$ for some $\alpha > 0$, then $\tilde{\Lambda}^* = \alpha^{-1} \mathbb{Z}^{1 \times n} \tilde{\mathcal{A}}^{-1}$ and

$$\epsilon_{\alpha \mathbb{Z}^{1 \times n} \tilde{\mathcal{A}}}(\sigma) = \sum_{z \in \mathbb{Z}^n \setminus \{0\}} \exp\left( -\left(\frac{\sigma}{\alpha}\right)^2 \cdot 2\pi^2 \|z \tilde{\mathcal{A}}^{-1}\|^2 \right). \tag{26}$$

We therefore have that for any fixed $\sigma, \tilde{\alpha}_1, \ldots, \tilde{\alpha}_n > 0$

$$\lim_{\alpha \to 0} \epsilon_{\alpha \mathbb{Z}^{1 \times n} \tilde{\mathcal{A}}}(\sigma) = 0. \tag{27}$$

**Statistics of the quantizer's input**

We prove the following.

**Lemma B.1.** *Let $W \sim \mathcal{N}(0, \sigma^2 I)$ be a Gaussian vector in $\mathbb{R}^{1 \times n}$, $L \in \mathbb{R}^{n \times n}$ be a lower triangular matrix, and $\tilde{\mathcal{A}} = \mathrm{diag}(\tilde{\alpha}_1, \ldots, \tilde{\alpha}_n) \in \mathbb{R}_+^{n \times n}$. Assume we apply Algorithm 1 with $Y = WL$, $L$ and $\mathcal{A} = \alpha \tilde{\mathcal{A}}$, for some $\alpha > 0$. Then*

$$Z_{\mathrm{SIC},1,k} = \mathrm{round}\left(\frac{W_k}{\alpha \tilde{\alpha}_k} + e_k\right), \tag{28}$$

*where $e_k$ is statistically independent of $W_k$ and satisfies $|e_k| < C(L, \mathcal{A}, n)$ with probability 1, where $0 < C(L, \mathcal{A}, n) < \infty$ is independent of $\alpha$.*

*Proof.* Let $Z_{\mathrm{SIC}}$ be the output of the algorithm, and let $\hat{Y} = Z_{\mathrm{SIC}} \mathcal{A} L$, and let $e_{\mathrm{SIC}} = Y - \hat{Y}$. As we have seen in Lemma 3.2, $e_{\mathrm{SIC}} \in \mathrm{CUBE} \cdot \mathcal{A} \cdot \mathrm{diag} \, L$. Inspecting the algorithm's successive rounding procedure, we have that

$$Z_{\mathrm{SIC}} = \mathrm{round}\left((Y - Z_{\mathrm{SIC}} \mathcal{A}(L - \mathrm{diag}(L))) \cdot (\mathcal{A} \, \mathrm{diag}(L))^{-1}\right) \tag{29}$$

$$= \mathrm{round}\left((Y - Z_{\mathrm{SIC}} \mathcal{A} L \cdot L^{-1}(L - \mathrm{diag}(L))) \cdot (\mathcal{A} \, \mathrm{diag}(L))^{-1}\right) \tag{30}$$

$$= \mathrm{round}\left((Y - \hat{Y} \cdot (I - L^{-1} \, \mathrm{diag}(L))) \cdot (\mathcal{A} \, \mathrm{diag}(L))^{-1}\right) \tag{31}$$

$$= \mathrm{round}\left((Y - (Y - e_{\mathrm{SIC}}) \cdot (I - L^{-1} \, \mathrm{diag}(L))) \cdot (\mathcal{A} \, \mathrm{diag}(L))^{-1}\right) \tag{32}$$

$$= \mathrm{round}\left((YL^{-1} \, \mathrm{diag}(L) + e_{\mathrm{SIC}} \cdot (I - L^{-1} \, \mathrm{diag}(L))) \cdot (\mathcal{A} \, \mathrm{diag}(L))^{-1}\right) \tag{33}$$

$$= \mathrm{round}\left((W \, \mathrm{diag}(L) + e_{\mathrm{SIC}} \cdot (I - L^{-1} \, \mathrm{diag}(L))) \cdot (\mathcal{A} \, \mathrm{diag}(L))^{-1}\right) \tag{34}$$

$$= \mathrm{round}\left(W \mathcal{A}^{-1} + e_{\mathrm{SIC}} \cdot (I - L^{-1} \, \mathrm{diag}(L)) \cdot (\mathcal{A} \, \mathrm{diag}(L))^{-1}\right). \tag{35}$$

We are left with characterizing the random vector

$$e = e_{\mathrm{SIC}} \cdot \tilde{L} \cdot (\mathcal{A} \, \mathrm{diag}(L))^{-1} \tag{36}$$

where we defined the strictly lower triangular matrix $\tilde{L} = I - L^{-1} \, \mathrm{diag} \, L$. First, since $\tilde{L}$ is lower triangular, $e_k$ is a deterministic function of $e_{\mathrm{SIC}}(k + 1 : n)$. Furthermore, by definition of the ZSIC algorithm $e_{\mathrm{SIC}}(k + 1 : n)$ is a deterministic function of $Y(k + 1 : n)$, and $Y(k + 1 : n)$ in turn, is a deterministic function of $W(k + 1 : n)$ since $L$ is lower triangular. Since $W_k \perp\!\!\!\perp W(k + 1 : n)$, it therefore follows that $e_k \perp\!\!\!\perp W_k$ as claimed. It remains to upper bound $|e_k|$. To that end, let $S = e_{\mathrm{SIC}} \cdot (\mathcal{A} \, \mathrm{diag}(L))^{-1}$ and note that $S \in \mathrm{CUBE}$ since $e_{\mathrm{SIC}} \in \mathrm{CUBE} \cdot \mathcal{A} \cdot \mathrm{diag} \, L$. We therefore have that

$$e = S(\tilde{\mathcal{A}} \cdot \mathrm{diag} \, L) \tilde{L} (\tilde{\mathcal{A}} \, \mathrm{diag}(L))^{-1}. \tag{37}$$

is independent of $\alpha$ and has bounded support. $\qquad \square$

### B.2. The proof

Let us jointly analyze both the Huffman-GPTQ/GPTQ algorithm and the PlainWaterSIC algorithm. In both cases the spacing matrix will be of the form

$$\mathcal{A} = \alpha \tilde{\mathcal{A}}, \tag{38}$$

where $\tilde{\mathcal{A}} = I$ for GPTQ, whereas for PlainWaterSIC it will be

$$\tilde{\mathcal{A}} = \mathrm{diag}\left(\tilde{\alpha}_1, \ldots, \tilde{\alpha}_n\right), \quad \text{where } \tilde{\alpha}_i = \frac{|L|^{1/n}}{|\ell_{ii}|} \quad i = 1, \ldots, n. \tag{39}$$

In both cases $|\tilde{\mathcal{A}}| = 1$, and the density of the corresponding lattice is controlled only by the parameter $\alpha$. We will show that

$$\lim_{\alpha \to 0} \frac{D(\alpha)}{\alpha^2} = \frac{1}{n} \frac{1}{12} \sum_{i=1}^{n} (\tilde{\alpha}_i \ell_{ii})^2. \tag{40}$$

and that

$$\lim_{\alpha \to 0}\left[H(Z_{\mathrm{SIC}}, i) - \frac{1}{2}\log(\alpha^2)\right] = \frac{1}{2}\log(2\pi e \sigma_W^2) - \log(\tilde{\alpha}_i), \quad i = 1, \ldots, n. \tag{41}$$

The expected rate $R(\alpha)$ is the result of applying entropy coding to each column $Z_{\text{SIC},:,i}$, $i = 1, \ldots, n$ of $Z_{\text{SIC}}$. Since rows of $W$ are independent, so are the $a$ entries in each column $Z_{\text{SIC},:,i}$ of $Z_{\text{SIC}}$. If $a$ is sufficiently large, we can apply universal entropy coding on the $i$th column, $i = 1, \ldots, n$, and the expected number of bits will be $a(H(Z_{\text{SIC},i}) + o(1))$, where $o(1)$ vanishes with $a$. We ignore this term in the remainder of the analysis since we take limit of $a \to \infty$. From (41) it therefore immediately follows that

$$\lim_{\alpha \to 0}[R(\alpha) - \frac{1}{2}\log(\alpha^2)] = \frac{1}{n}\sum_{i=1}^{n}\lim_{\alpha \to 0}\left[H(Z_{\text{SIC}}, i) - \frac{1}{2}\log(\alpha^2)\right]$$
$$= \frac{1}{2}\log(2\pi e \sigma_W^2) - \frac{1}{n}\log|\tilde{\mathcal{A}}|$$
$$= \frac{1}{2}\log(2\pi e \sigma_W^2), \tag{42}$$

where the last equality follows from our assumption that $|\tilde{\mathcal{A}}| = 1$. Combining (40) and (42) we obtain

$$\lim_{\alpha \to 0}\left[R(\alpha) + \frac{1}{2}\log\left(D(\alpha)\right)\right] = \frac{1}{2}\log\left(\frac{2\pi e}{12}\frac{1}{n}\sum_{i=1}^{n}(\tilde{\alpha}_i \ell_{ii})^2 \sigma_W^2\right). \tag{43}$$

Subtracting $R_{\text{WF}}(D(\alpha), \Sigma_X) = R_{\text{High-Rate}}(D(\alpha), \Sigma)$ (since $D(\alpha) \to 0$ as $\alpha \to 0$) from both sides, and recalling that

$$|\Sigma_X|^{1/n} = |L|^{2/n} = |L\tilde{\mathcal{A}}|^{2/n} = \prod_{i=1}^{n}(\tilde{\alpha}_i \ell_{ii})^{2/n},$$

since $|\tilde{\mathcal{A}}| = 1$, we obtain

$$\lim_{\alpha \to 0}[R(\alpha) - R_{\text{WF}}(D(\alpha), \Sigma_X)] = \frac{1}{2}\log\left(\frac{2\pi e}{12}\frac{\frac{1}{n}\sum_{i=1}^{n}(\tilde{\alpha}_i \ell_{ii})^2}{\prod_{i=1}^{n}(\tilde{\alpha}_i \ell_{ii})^{2/n}}\right). \tag{44}$$

The claimed result now follows from substituting the corresponding $\tilde{\alpha}_1, \ldots, \tilde{\alpha}_n$ for GPTQ and for WaterSIC. It therefore only remains to prove (40) and (41).

**Proof of** (40). Consider the lattice $\Lambda = \alpha\mathbb{Z}^{1 \times n}\tilde{\mathcal{A}}$. Recall that $e_{\text{SIC}} = WL - Z_{\text{SIC}}\alpha\tilde{\mathcal{A}}L$ and that $e_{\text{SIC}} \in \alpha\text{CUBE}\cdot\tilde{\mathcal{A}}\text{diag}(L)$. Therefore,

$$e_{\text{SIC}} = e_W L \tag{45}$$

where

$$e_W = W - \alpha Z_{\text{SIC}}\tilde{\mathcal{A}}, \tag{46}$$

and $Z_{\text{SIC}} \in \mathbb{Z}^{1 \times n}$ is chosen such that $e_W \in \alpha\text{CUBE} \cdot \tilde{\mathcal{A}}\text{diag}(L)L^{-1} \triangleq \mathcal{P}$. We will show that $e_W$ is nearly uniform on $\mathcal{P}$, and it will then follow that so is $e_{\text{SIC}}$.

Let $f_{e_W}(x)$ denote the probability density function (pdf) of the random vector $e_W$. Note that

$$f_{e_W}(x) = f_{\sigma_W, \tilde{\Lambda}}(x)\mathbb{1}\{x \in \mathcal{P}\}, \tag{47}$$

where $\tilde{\Lambda} = \alpha\mathbb{Z}^{1 \times n}\tilde{\mathcal{A}}$. By definition of the flatness factor we have that

$$f_{e_W}(x) \in [1 \pm \epsilon_{\tilde{\Lambda}}(\sigma_W)]\frac{1}{\text{Vol}(\mathcal{P})}\mathbb{1}\{x \in \mathcal{P}\} \tag{48}$$

It therefore follows that the pdf of $e_{\text{SIC}} = e_W L$ satisfies

$$f_{e_{\text{SIC}}}(x) \in [1 \pm \epsilon_{\tilde{\Lambda}}(\sigma_W)]\frac{1}{\text{Vol}(\mathcal{P}L)}\mathbb{1}\{x \in \mathcal{P}L\}. \tag{49}$$

Consequently,

$$D = \frac{1}{n}\mathbb{E}\|e_{\text{SIC}}\|^2 = \int_{x \in \mathcal{P}L} \|x\|^2 f_{e_{\text{SIC}}}(x)dx$$

$$\in [1 \pm \epsilon_{\tilde{\Lambda}}(\sigma_W)]\,\mathbb{E}\|U_{\mathcal{P}L}\|^2 \tag{50}$$

$$\in [1 \pm \epsilon_{\tilde{\Lambda}}(\sigma_W)]\frac{1}{n}\frac{\alpha^2}{12}\sum_{i=1}^{n}(\tilde{\alpha}_i \ell_{ii})^2, \tag{51}$$

where $U_{\mathcal{P}L} \sim \text{Uniform}(\mathcal{P}L)$. By (27) we have that $\lim_{\alpha \to 0} \epsilon_{\tilde{\Lambda}}(\sigma_W) = 0$ and (40) indeed holds.

**Proof of** (41). By Lemma B.1 we have

$$Z_{\text{SIC},i} = \text{round}\left(\frac{1}{\alpha}V_\alpha\right), \tag{52}$$

where

$$V_\alpha = \frac{W_i}{\tilde{\alpha}_k} + \alpha e_i. \tag{53}$$

Here $W_i \sim \mathcal{N}(0, \sigma_W^2)$, and $e_i$ is bounded in an interval independent of $\alpha$ with probability 1, and is statistically independent of $W_i$. For $\Delta > 0$ and a random variable $X$ let $X^\Delta = \Delta \cdot \text{round}\left(\frac{X}{\Delta}\right)$. A classic result of Rényi (Rényi, 1959, Theorem 1), (Polyanskiy & Wu, 2024, eq. 2.21) shows that if $h(X)$ exists and $H(\text{round}(X)) < \infty$ then

$$\lim_{M \to \infty} H(X^{1/M}) - \log M = h(X) + E(M, P_X), \tag{54}$$

where $E(M, P_X)$ vanishes with $M$. Let $\mathcal{V}_{\alpha_0}$ be the family of distributions on $V_\alpha$ for all $\alpha < \alpha_0$. Inspecting the proof of (Rényi, 1959, Theorem 1), we see that for sufficiently small $\alpha_0$ his equations (40-41) can be made to hold simultaneously with the same $L(\varepsilon)$ for all distributions in $\mathcal{V}_{\alpha_0}$. It therefore follows that $E(M, P_X)$ converges to 0 with $M \to \infty$ uniformly for all $P_X \in \mathcal{V}_{\alpha_0}$. Since $H(\text{round}(\frac{1}{\alpha} \cdot V_\alpha) = H(V_\alpha^\alpha)$ we have

$$\lim_{M \to \infty} \left[H(Z_{\text{SIC},i}) - \log(\alpha)\right]\bigg|_{\alpha=1/M} = \lim_{M \to \infty} h(V_{1/M}). \tag{55}$$

Finally, using the continuity of entropy, specifically (Linder & Zamir, 1994) that for $e \perp\!\!\!\perp X$ where $\mathbb{E}(e^2) < \infty$ and $h(X)$ exists it holds that

$$\lim_{\alpha \to 0}[h(X + \alpha e)] = h(X), \tag{56}$$

we obtain

$$\lim_{\alpha \to 0}\left[H(Z_{\text{SIC},i}) - \log(\alpha)\right] = \frac{1}{2}\log(2\pi e \sigma_W^2) - \log(\tilde{\alpha}_i), \tag{57}$$

as claimed.

## C. Calibration statistics and adaptive mixing

**Collecting raw calibration data.** For calibration and tuning, we use the full WikiText-2 training split. We concatenate the text into a single token stream (with a single BOS token prepended) and partition it into non-overlapping sequences of length 2048, discarding any remainder. The exact number of sequences depends on the tokenizer (e.g., $\approx 1189$ for Llama-3.2-1B). Only the first sequence begins with a BOS token; subsequent sequences start at arbitrary token boundaries. We verified on Llama-3.2-1B that prepending a BOS token to *every* sequence affects neither perplexity nor the resulting calibration statistics.

Given these sequences, and assuming that preceding layers are already quantized, we run both the unquantized and (partially) quantized models to collect the layer inputs $X$ and $\hat{X}$, as well as the residual stream states $R$ and $\hat{R}$. We then form the

---

**Algorithm 3** WATERSIC: full algorithm details

---

**Require:** Weight matrix $W \in \mathbb{R}^{a \times n}$, covariance $\Sigma_X \in \mathbb{R}^{n \times n}$, scale parameter $c > 0$, damping $\delta \geq 0$.
**Require:** Covariance $\Sigma_{\hat{X}}, \Sigma_{X,\hat{X}}, \Sigma_{\Delta,\hat{X}}$          ▷ If not available, set first two to $\Sigma_X$ and the last one to 0.
**Ensure:** Reconstructed weights $\hat{W}$, effective rate $R_{\text{eff}}$

    **Phase 1: Setup**
1:   $H \leftarrow \Sigma_{\hat{X}} + \delta \cdot \text{mean}(\text{diag}(\Sigma_{\hat{X}})) \cdot I_n$          ▷ For stability in early layers
2:   $L \leftarrow \text{chol}(H)$          ▷ Lower-triangular: $H = LL^\top$
3:   $Y \leftarrow (W \Sigma_{X,\hat{X}} + \Sigma_{\Delta,\hat{X}}) (L^\top)^{-1}$          ▷ Use triangular solver for efficiency
4:   $\alpha_k \leftarrow c/L_{k,k}$ for $k = 1, \dots, n$

    **Phase 2: ZSIC with LMMSE correction**
5:   **for** $i = n, n-1, \dots, 1$ **do**
6:      $\mathbf{z}_i \leftarrow \text{round}(Y_{:,i} / c)$
7:      $\gamma_i \leftarrow \mathbf{z}_i^\top Y_{:,i} / (c \|\mathbf{z}_k\|^2)$
8:      $Y \leftarrow Y - \gamma_k \alpha_k \mathbf{z}_k \mathbf{1}_k^\top$          ▷ $\mathbf{1}_k^\top$: row $k$ of $L$
9:   **end for**

    **Phase 3: Rate computation**
10: $\mathcal{H} \leftarrow -\sum_v p_v \log_2 p_v, \quad p_v = |\{(i,k) : Z_{ik} = v\}|/(an)$
11: $R_{\text{eff}} \leftarrow \mathcal{H} + 16/a + 16/n$          ▷ Entropy + side-info overhead

    **Phase 4: Diagonal rescaler optimization**
12: $\hat{W}_0 \leftarrow Z \, \text{diag}(\alpha)$
13: $T, \Gamma \leftarrow \text{FINDOPTIMALRESCALERS}(\hat{W}_0, W, \Sigma_X, \Sigma_{\hat{X}}, \Sigma_{X,\hat{X}}, \Sigma_{\Delta,\hat{X}}; \gamma_{\text{init}} = \gamma)$
14: $\hat{W} \leftarrow T Z \Gamma \, \text{diag}(\alpha)$          ▷ $T = \text{diag}(t)$, $\Gamma = \text{diag}(\tilde{\gamma})$
15: **return** $R_{\text{eff}}, \hat{W}$

---

required (cross-)covariance estimates by averaging over all token positions (and their attention-weighted counterparts, where applicable).

**Selecting adaptive mixing coefficients.** As described in (58) for some layers we undertake additional optimization in the form of mixing unquanized statistics into quantized one.

Specifically, first we introduce a mixing parameter $\epsilon_{\text{qr}} \in [0, 1]$ that interpolates between the drift-corrected and original statistics:

$$\Sigma_{\hat{X}}^{(\text{mix})} = (1 - \epsilon_{\text{qr}}) \Sigma_{\hat{X}} + \epsilon_{\text{qr}} \Sigma_X,$$
$$\Sigma_{X,\hat{X}}^{(\text{mix})} = (1 - \epsilon_{\text{qr}}) \Sigma_{X,\hat{X}} + \epsilon_{\text{qr}} \Sigma_X. \tag{58}$$

Setting $\epsilon_{\text{qr}} = 0$ recovers full drift correction (16), while $\epsilon_{\text{qr}} = 1$ falls back to the original (unquantized) Hessian.

Once the drift mixing is fixed, we apply attention weighting on top of the resulting covariances. Let $\Sigma_\bullet^{(w)}$ denote the attention-weighted estimate from (19) computed from the mixed statistics $\Sigma_\bullet^{(\text{mix})}$. A second parameter $\epsilon_{\text{aw}} \in [0, 1]$ interpolates between the weighted and uniform versions:

$$\Sigma_\bullet^{(\text{final})} := (1 - \epsilon_{\text{aw}}) \Sigma_\bullet^{(w)} + \epsilon_{\text{aw}} \Sigma_\bullet^{(\text{mix})},$$
$$\Sigma_\bullet \in \{\Sigma_X, \Sigma_{\hat{X}}, \Sigma_{X,\hat{X}}\}. \tag{59}$$

Thus, the attention importance scores weigh the already drift-mixed covariances, and $\epsilon_{\text{aw}}$ controls how strongly this reweighting is applied.

We optimize $(\epsilon_{\text{qr}}, \epsilon_{\text{aw}})$ per layer via a lightweight coordinate search. For a given layer, let $\hat{w}_q, \hat{w}_k, \hat{w}_v$ denote the quantized projections obtained using the blended statistics (58)-(59). We minimize the relative MSE of the input to $w_o$, i.e., the output

---

**Algorithm 4** FINDOPTIMALRESCALERS: Alternating optimization of diagonal row and column rescalers

---

**Require:** $\hat{W}_0 \in \mathbb{R}^{a \times n}$ (pre-rescaler reconstruction), $W \in \mathbb{R}^{a \times n}$ (original weights), $\Sigma_X, \Sigma_{\hat{X}}, \Sigma_{X,\hat{X}}, \Sigma_{\Delta,\hat{X}} \in \mathbb{R}^{n \times n}$, initial
$\quad \gamma^{(0)} \in \mathbb{R}^n$, tolerance $\varepsilon$, ridge $\lambda \geq 0$

**Ensure:** Diagonal matrices $T = \mathrm{diag}(t), \Gamma = \mathrm{diag}(\gamma)$

$\quad$ // **Objective:** $\mathcal{J}(T, \Gamma) = \frac{1}{an} \mathrm{tr}\big(W \Sigma_X W^\top - 2\,(W\Sigma_{X,\hat{X}} + \Sigma_{\Delta,\hat{X}})(T\hat{W}_0\Gamma)^\top + T\hat{W}_0\Gamma\,\Sigma_{\hat{X}}\,\Gamma\hat{W}_0^\top T\big)$

1: $t \leftarrow \mathbf{1}_a; \quad \gamma \leftarrow \gamma^{(0)}$
2: $s \leftarrow \|t\|_1 / a; \quad t \leftarrow t/s; \quad \gamma \leftarrow s\,\gamma$ $\qquad\qquad\qquad\qquad\qquad\qquad\qquad$ ▷ Normalize: $\|t\|_1 = a$
3: $\mathcal{L}_{\mathrm{prev}} \leftarrow \mathcal{J}(\mathrm{diag}(t), \mathrm{diag}(\gamma))$

4: **for** iter $= 1, 2, \ldots$ **do**
$\quad$ $\Gamma$**-step:**
5: $\quad F \leftarrow \hat{W}_0^\top \mathrm{diag}(t^2)\hat{W}_0$ $\qquad\qquad\qquad\qquad\qquad\qquad\qquad\qquad\qquad\qquad\qquad$ ▷ $n \times n$
6: $\quad G \leftarrow \Sigma_{\hat{X}} \odot F$ $\qquad\qquad\qquad\qquad\qquad\qquad\qquad\qquad\qquad$ ▷ Hadamard product, $n \times n$
7: $\quad \mathbf{d} \leftarrow \mathrm{diag}\big(\hat{W}_0^\top \mathrm{diag}(t)\,(W\Sigma_{X,\hat{X}} + \Sigma_{\Delta,\hat{X}})\big)$ $\qquad\qquad\qquad\qquad\qquad$ ▷ $n$-vector
8: $\quad \gamma \leftarrow (G + \lambda I_n)^{-1}\mathbf{d}$

$\quad$ $T$**-step:**
9: $\quad \mathbf{p} \leftarrow \mathrm{diag}\big((W\Sigma_{X,\hat{X}} + \Sigma_{\Delta,\hat{X}})\,\mathrm{diag}(\gamma)\,\hat{W}_0^\top\big)$ $\qquad\qquad\qquad\qquad\qquad$ ▷ $a$-vector
10: $\quad \mathbf{q} \leftarrow \mathrm{diag}\big(\hat{W}_0\,\mathrm{diag}(\gamma)\,\Sigma_{\hat{X}}\,\mathrm{diag}(\gamma)\,\hat{W}_0^\top\big)$ $\qquad\qquad\qquad\qquad\qquad$ ▷ $a$-vector
11: $\quad t_i \leftarrow p_i / (q_i + \lambda)$ for $i = 1, \ldots, a$

$\quad$ **Re-normalize and check convergence:**
12: $\quad s \leftarrow \|t\|_1 / a; \quad t \leftarrow t/s; \quad \gamma \leftarrow s\,\gamma$
13: $\quad \mathcal{L}_{\mathrm{curr}} \leftarrow \mathcal{J}(\mathrm{diag}(t), \mathrm{diag}(\gamma))$
14: $\quad$ **if** $|\mathcal{L}_{\mathrm{curr}} - \mathcal{L}_{\mathrm{prev}}| / (|\mathcal{L}_{\mathrm{prev}}| + 10^{-12}) < \varepsilon$ **then**
15: $\qquad$ **break**
16: $\quad$ **end if**
17: $\quad \mathcal{L}_{\mathrm{prev}} \leftarrow \mathcal{L}_{\mathrm{curr}}$
18: **end for**
19: **return** $T = \mathrm{diag}(t), \quad \Gamma = \mathrm{diag}(\gamma)$

---

of the multi-head attention block before the output projection:

$$\min_{\epsilon_{\mathrm{qr}}, \epsilon_{\mathrm{aw}} \in [0,1]} \frac{\mathbb{E}\left[\left\|\mathrm{Attn}(X; w_q, w_k, w_v) - \mathrm{Attn}(\hat{X}; \hat{w}_q, \hat{w}_k, \hat{w}_v)\right\|_F^2\right]}{\mathbb{E}\left[\|\mathrm{Attn}(X; w_q, w_k, w_v)\|_F^2\right]}, \tag{60}$$

where $\mathrm{Attn}(X; w_q, w_k, w_v)$ denotes the multi-head self-attention output (including the softmax) given input activations $X$ and projection weights, and $\hat{X}$ are the (already quantized) activations produced by the preceding layers. Measuring distortion at the $w_o$ input rather than at individual projection outputs captures the error amplification through the softmax nonlinearity, which empirically dominates the quantization loss in attention layers.

The objective in (60) is empirically unimodal in each coordinate when the other is held fixed. We therefore first optimize $\epsilon_{\mathrm{qr}}$ with $\epsilon_{\mathrm{aw}}$ at a default value using golden-section search over $[0, 1]$, then optimize $\epsilon_{\mathrm{aw}}$ with $\epsilon_{\mathrm{qr}}$ fixed at its optimum. Each evaluation re-quantizes $(w_q, w_k, w_v)$ jointly and performs a forward pass through the attention block on the calibration set. The resulting parameters $(\epsilon_{\mathrm{qr}}^\star, \epsilon_{\mathrm{aw}}^\star)$ are stored per layer and used during the final quantization pass.

**Hessian damping.** It is a common practice starting from GPTQ codebase to replace

$$\Sigma_X \leftarrow \Sigma_X + \delta I_n \,,$$

where $\delta$ is a hyperparameter determining regularization strength of the collected covariance matrix (by default, $\delta = \frac{0.1}{n} \mathrm{tr}\,\Sigma_X$). Note that adding this term is equivalent to replacing loss objective by

$$\min_{\hat{W}} \mathrm{tr}(W - \hat{W})\Sigma_X(W - \hat{W})^\top + \delta\|W - \hat{W}\|_F^2 \,,$$

which evidently safeguards $\hat{W}$ from deviating from $W$ too much.

Thus, in the presence of drift correction and residual compensation we are to minimize objective:

$$\min_{\hat{W}} \mathbb{E}[\|WX + R - (\hat{W}\hat{X} + \hat{R})\|_2^2] + \delta\|W - \hat{W}\|_F^2,$$

which is equivalent to modifying

$$\Sigma_X \leftarrow \Sigma_X + \delta I_n$$
$$\Sigma_{\hat{X}} \leftarrow \Sigma_{\hat{X}} + \delta I_n$$
$$\Sigma_{X,\hat{X}} \leftarrow \Sigma_{X,\hat{X}} + \delta I_n \qquad \text{(note!)}$$
$$\Sigma_{\Delta,\hat{X}} \leftarrow \Sigma_{\Delta,\hat{X}} \qquad \text{(not a typo!)}$$

Damping $\delta I$ is applied after the adaptive mixing, on the resulting blended matrices.

## D. Hyperparameters

Here we summarize the hyperparameters used in our experiments and describe how each plot was produced.

**Common settings.** We quantized layers sequentially, collecting statistics of $(X, \hat{X}, R, \hat{R})$ at the input to each subsequent layer. The reported rate is the parameter-count-weighted average of the per-layer rates.

For evaluation, we used the test split of WikiText-2 and computed perplexity (PPL) with a context window of 2048. In addition to PPL, we computed KL divergence and several reasoning benchmarks, including MMLU and HellaSwag.

We evaluated WaterSIC and GPTQ on Llama-3.2-1B, Qwen3-8B, Llama-3-8B, Llama-2-7B and Llama-3-70B.

**GPTQ.** We used $\delta = 0.1$ (the default) for all models except Llama-3-8B, where we found that $\delta = 0.01$ performed significantly better at some rates. We evaluated with $\texttt{groupsize} = -1$, $\texttt{blocksize} = 128$, and $\texttt{actorder} = \texttt{False}$. To obtain different target rates, we varied the $\texttt{maxq}$ parameter and computed the entropy of the resulting integer matrices.

As input to the algorithm, we used quantized activation statistics $\hat{X}$, as they consistently produced better results. In the main plots, this variant is labeled as Huffman-GPTQ (HPTQ), following (Chen et al., 2026). Entries labeled GPTQ, on the other hand, correspond to the same algorithm with the rate set to $\log_2(\texttt{maxq})$ (i.e., without entropy coding). For Qwen3-8B, we did not run Huffman-GPTQ ourselves; instead, we report the results from (Chen et al., 2026).

**WaterSIC.** With dead-feature erasure enabled, we found that very small damping ($\delta = 10^{-4}$) is sufficient, in contrast to the much larger damping (e.g., $\approx 10\%$) commonly used in standard GPTQ.

For each layer, we use a secant method to find the value of $c$ in Algorithm 3 that yields the target rate. We exploit the approximately linear relationship between $c$ and the resulting entropy: an initial evaluation at the target rate reveals an offset, and a secant correction refines the estimate in 2–3 iterations to within $< 0.005$ bits of the target. To reduce computation, this search compresses only a randomly sampled $10\%$ of the rows of $W$. After convergence, we rerun the algorithm with the selected value of $c$ on the full matrix to produce the final quantized weights.

As we quantize the model sequentially, we maintain a running rate budget initialized to the global target. At each step, we allocate this remaining budget evenly across the unquantized matrices and calibrate the current layer to match its assigned share. This procedure keeps the final average rate extremely close to the requested target. In addition, because dead-feature erasure reduces the effective dimensionality (and thus the rate) of some early layers, the leftover budget is redistributed to later layers, resulting in a mild increase in per-layer rates toward the end of the network.

Our best results across models used activation drift correction (16) and residual stream compensation (18), together with weighted calibration (19) and adaptive mixing (58) for joint quantization of the $(w_q, w_k, w_v)$ projections. We found that 10 iterations of golden-section search provide sufficient precision for each mixing parameter. The per-layer procedure is:

1. **Drift-mixing optimization.** Holding $\epsilon_{\text{aw}} = 0$ fixed, run golden-section search over $\epsilon_{\text{qr}} \in [0, 1]$. At each candidate value, re-quantize $(w_q, w_k, w_v)$ jointly (each time finding the appropriate $c$ via the secant method to match the target rate) and evaluate the relative MSE at the $w_o$ input over the calibration set. Select $\epsilon_{\text{qr}}^{\star}$ minimizing (60).

2. **Attention-weighting optimization.** Fixing $\epsilon_{qr} = \epsilon_{qr}^{\star}$, run golden-section search over $\epsilon_{aw} \in [0, 1]$ using the same re-quantize-and-evaluate procedure, yielding $\epsilon_{aw}^{\star}$.

3. **Final quantization.** With $(\epsilon_{qr}^{\star}, \epsilon_{aw}^{\star})$ fixed, quantize $(w_q, w_k, w_v)$ at the target rate assigned by the global rate controller.

Each golden-section iteration requantizes three matrices for a single layer (with the secant method determining the scale $c$ at each evaluation point) and runs a forward pass through the attention block on the calibration set. Because the secant method converges in 2–3 iterations, rate-matching each evaluation point adds minimal overhead compared to the quantization itself.

**Qwen3-8B.** For this model, we identified a small number of outlier rows in the FFN gate and up-projection matrices that negatively impact quantization stability. Specifically, rows 5723 and 8518 in layer 6 $(w_1, w_3)$ and rows 2271 and 1875 in layer 16 $(w_1, w_3)$ have anomalously large norms. These outliers inflate the effective scale of the corresponding down-projection $w_2$ (which must compensate for their magnitude), leading to degraded quantization fidelity. We verified that zeroing these rows in the *unquantized* model increases WikiText-2 test perplexity by less than $0.01$, indicating that they encode near-degenerate features rather than useful signal. We therefore zero these rows in both the unquantized and quantized models prior to quantization and, correspondingly, zero the associated columns in $w_2$. This removes a source of numerical instability without a meaningful loss in model quality, and substantially reduces quantization error in the affected layers.

Additionally, on Qwen3-8B we observed a sharp transition around layer 16: the relative MSE at the input to the layer-16 $w_2$ matrix is substantially higher than in earlier layers. After this point, the coordinate search increasingly prefers the unquantized statistics over the quantized-model ones: as shown in Table 3, the optimal drift-mixing coefficients $\epsilon_{qr}^{\star}$ are close to 1 for most subsequent layers (across all three rates). In contrast, the attention-weight mixing parameter $\epsilon_{aw}^{\star}$ remains at 0 (full attention weighting) for the majority of layers and becomes nonzero only in a subset of deeper layers (Table 4). We hypothesize that beyond this depth, quantization noise is strongly amplified, which degrades the quality of the estimated quantized-model covariances and makes activation drift compensation less effective in practice.

*Table 3.* Optimal drift-mixing coefficients $\epsilon_{qr}^{\star}$ selected by adaptive mixing for Qwen3-8B. Values are reported per layer for three target rates (bits per parameter). Larger $\epsilon_{qr}^{\star}$ indicates a stronger preference for the unquantized statistics $\Sigma_X$ over the drift-corrected statistics $(\Sigma_{\hat{X}}, \Sigma_{X,\hat{X}})$. Note a "phase change" at layer 15. Calibration at CTX=2048.

| Layer | 2.125 | 3.125 | 4.125 | Layer | 2.125 | 3.125 | 4.125 |
|-------|-------|-------|-------|-------|-------|-------|-------|
| 0 | 0.7641 | 0.5064 | 1.0000 | 18 | 0.9784 | 1.0000 | 0.9703 |
| 1 | 0.0132 | 0 | 0 | 19 | 1.0000 | 0.9786 | 0.9913 |
| 2 | 0.0132 | 0.0016 | 0 | 20 | 1.0000 | 1.0000 | 1.0000 |
| 3 | 0 | 0.0201 | 0.0098 | 21 | 0.9657 | 0.9874 | 1.0000 |
| 4 | 0.1459 | 0 | 0.0081 | 22 | 0.9956 | 1.0000 | 1.0000 |
| 5 | 0.0545 | 0 | 0.0047 | 23 | 1.0000 | 0.9987 | 1.0000 |
| 6 | 0 | 0.0473 | 0.0900 | 24 | 0.9951 | 0.9917 | 1.0000 |
| 7 | 0.2353 | 0.2905 | 0.1459 | 25 | 0.9894 | 0.9951 | 0.9951 |
| 8 | 0.1462 | 0.3824 | 0 | 26 | 0.9820 | 0.9874 | 0.9542 |
| 9 | 0.2844 | 0.0803 | 0.2937 | 27 | 0.9874 | 0.9820 | 0.9870 |
| 10 | 0.3262 | 0.0701 | 0.1653 | 28 | 0.9561 | 1.0000 | 0.9755 |
| 11 | 0 | 0 | 0 | 29 | 0.9166 | 0.9820 | 0.9738 |
| 12 | 0.3820 | 0.1834 | 0.2918 | 30 | 0.7639 | 1.0000 | 0.9868 |
| 13 | 0.8204 | 0.3851 | 0.4719 | 31 | 1.0000 | 1.0000 | 0.9855 |
| 14 | 0.7685 | 0.2535 | 0.4427 | 32 | 1.0000 | 0.9525 | 0.9836 |
| 15 | 0.9849 | 0.9392 | 0.9230 | 33 | 1.0000 | 0.9917 | 1.0000 |
| 16 | 0.9950 | 1.0000 | 0.9098 | 34 | 1.0000 | 1.0000 | 1.0000 |
| 17 | 0.9675 | 1.0000 | 0.9769 | 35 | 0.8198 | 0.8404 | 0.8968 |

*Table 4.* Optimal attention-weight mixing coefficients $\epsilon_{aw}^{\star}$ selected by adaptive mixing for Qwen3-8B. Values are reported per layer for three target rates (bits per parameter). Here $\epsilon_{aw}^{\star} = 0$ corresponds to full attention-weighted calibration, while larger values interpolate toward the uniformly weighted covariances. Calibration at CTX=2048.

| Layer | 2.125 | 3.125 | 4.125 | Layer | 2.125 | 3.125 | 4.125 |
|-------|-------|-------|-------|-------|-------|-------|-------|
| 0 | 0.5197 | 0.4721 | 0.5492 | 18 | 0 | 0 | 0 |
| 1 | 0 | 0 | 0 | 19 | 0.3839 | 0.6180 | 0.4114 |
| 2 | 0 | 0 | 0 | 20 | 0 | 0 | 0 |
| 3 | 0 | 0 | 0 | 21 | 1.0000 | 0.4752 | 0.2012 |
| 4 | 0 | 0 | 0 | 22 | 0.2310 | 0.4191 | 0.4047 |
| 5 | 0 | 0 | 0 | 23 | 0.2710 | 1.0000 | 0.3731 |
| 6 | 0 | 0 | 0 | 24 | 0.4346 | 0.2341 | 0.4741 |
| 7 | 0 | 0 | 0 | 25 | 0.3951 | 0.3820 | 0.7082 |
| 8 | 0 | 0 | 0 | 26 | 0.4769 | 0.3212 | 0.4702 |
| 9 | 0 | 0 | 0 | 27 | 0.3293 | 0.3820 | 0.7639 |
| 10 | 0 | 0 | 0 | 28 | 0.3607 | 0.7082 | 1.0000 |
| 11 | 0 | 0 | 0 | 29 | 0.5279 | 0.3731 | 0.6130 |
| 12 | 0 | 0 | 0 | 30 | 0 | 0 | 0 |
| 13 | 0 | 0 | 0 | 31 | 1.0000 | 0.3839 | 0.6180 |
| 14 | 0 | 0 | 0 | 32 | 0.3769 | 0.6161 | 0.0914 |
| 15 | 0 | 0 | 0 | 33 | 0.8448 | 0.6262 | 0.5798 |
| 16 | 0 | 0 | 0 | 34 | 0 | 0.2361 | 0.6130 |
| 17 | 0 | 0 | 0 | 35 | 0.0007 | 0.0027 | 0.0024 |

**WaterSIC-FT.** For post-quantization finetuning, we use AdamW without weight decay, with cosine annealing from a peak learning rate of $5 \times 10^{-4}$ to $5 \times 10^{-6}$ over 4 epochs on the full WikiText-2 training split (1189 sequences of length 2048). Gradient checkpointing is applied at every transformer block to keep activation memory at $O(1)$ blocks. The teacher's final hidden states (before the output head) are precomputed once and cached on CPU, eliminating the need to run the teacher forward pass at each training step and roughly halving per-step compute. During the forward pass, integer codes $Z$ are loaded from CPU to GPU per-layer on demand, so that peak GPU memory is dominated by a single block's activations

rather than the full quantized model. The rescaler parameters $t$ and $\gamma$ are rounded to half-precision via a straight-through estimator during training, so the optimizer adapts to the deployed (BF16-stored) parameter precision.

## E. Diagnostic plots and ablations

**Row-column rescalers.** On Fig. 4 we demonstrate how values of $T$ (row rescalers) and $\Gamma$ (column rescalers) change with rate. As expected theoretically, the LMMSE correction is indispensable for low-rate quantization and is not necessary for rates above $R \geq 4$ bit.

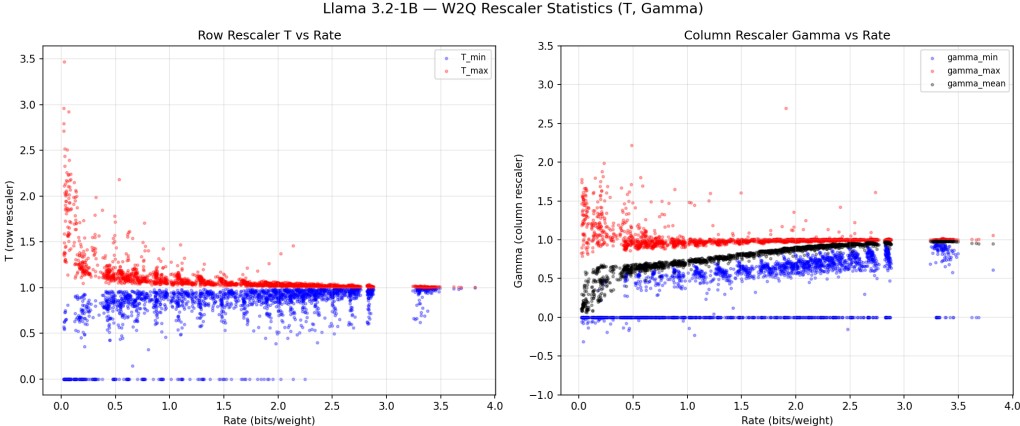

*Figure 4.* Diagonal rescalers statistics. Llama-3.2-1B, various rates. Row is the reduction (in-channel) dimension, so that row rescaler affects one out-channel, and column rescaler affects one in-channel. Calibration at CTX=2048.

**Zero coordinates and dead features.** One issue we encountered is that LayerNorm in Llama-3.2-1B effectively zeros out certain coordinates at layer inputs, which can make the empirical covariance matrices nearly singular. See Table 5 for the list of offending coordinates, defined as those with variance $< 10^{-4}$ times the mean variance across all $2048$ coordinates. This observation motivated the *dead feature erasure* heuristic described in Section 4: we remove near-zero-variance input dimensions before computing the Cholesky factor and optimizing rescalers, solve the reduced system, and then expand the quantized weight back to the original size by inserting zeros at the erased positions. In practice, this greatly improves numerical stability (both for the Cholesky decomposition and for rescaler optimization) while having negligible effect on reconstruction quality, since the erased coordinates carry little signal.

| Layer | Low-variance input indices |
|---|---|
| Layer 0, ATTN input | 278* |
| Layer 0, MLP input | 64*, 146*, 509, 1280*, 1618*, 1938*, 2023* |
| Layer 1, ATTN input | 146, 735, 762, 841, 894, 1002, 1314, 1334, 1476, 1503, 1619, 2037 |
| Layer 1, MLP input | 64*, 146*, 509, 1228, 2023 |
| Layer 2, ATTN input | 1159, 1314, 2023* |
| Layer 2, MLP input | 146*, 2023* |
| Layer 3, MLP input | 146, 2023 |
| Layer 4, MLP input | 146 |
| Layer 5, MLP input | 146 |

*Table 5.* Llama-3.2-1B: input features with standard deviation below 1% (asterisk) and 0.1% (no asterisk) of the mean standard deviation among all coordinates. These near-zero coordinates are produced by a diagonal multiplier inside the RMSnorm at the respective layer's input. Calibration at CTX=2048.

**Unequal rate.** Fig. 5 shows an example of the rate distribution among the different columns (in-channels) of matrices in one particular layer, as well the rate distribution among all the columns (of all layers) of Qwen3-8B, when compressing to target rate of 2.125 bits.

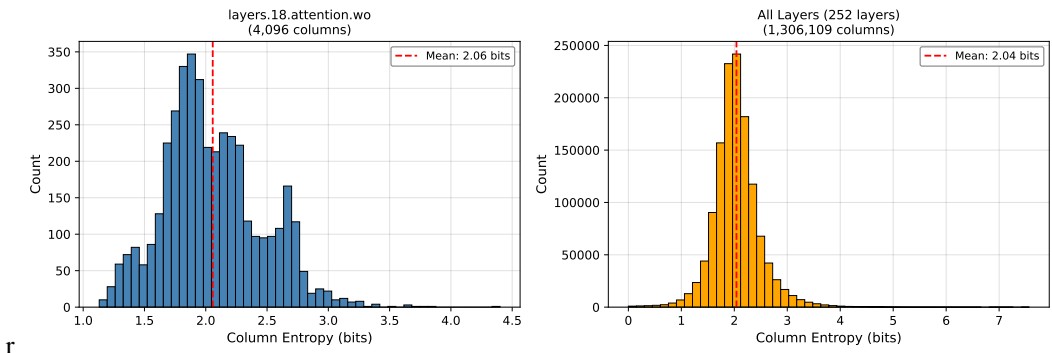

*Figure 5.* Distribution of actual compression rates per in-channel inside a single layer (left) and over all layers (right) for Qwen3-8B model. Calibration at CTX=2048.

Additionally, we study how well the entropy translates into algorithmically achievable compression. For each quantized weight matrix, we serialize the integer codes column-by-column (i.e., all entries sharing the same input feature are contiguous in the byte stream) and pack them into the smallest sufficient integer type (`int8` or `int16`). We then compress this byte stream with standard lossless codecs, including Zstandard (level 22) and LZMA (preset 9).

To this end, we evaluate several quantized layers of Llama-3.2-1B produced by our algorithm at a target rate of 2 bits per parameter. We report the entropy of all matrix entries, the maximum and average per-column entropies, along with the compression rates achieved by Zstandard and LZMA, in Table 6.

| Layer | Matrix | entropy (all matrix entries) | max(col-entropy) | avg(col-entropy) | zstd (bpp) | lzma (bpp) |
|-------|--------|------------------------------|------------------|------------------|------------|------------|
| 6 | attention.wk | 1.963 | 4.100 | 1.866 | 2.037 | 2.126 |
| | attention.wo | 1.975 | 4.397 | 1.853 | 2.016 | 2.107 |
| | attention.wq | 1.994 | 4.330 | 1.923 | 2.061 | 2.147 |
| | attention.wv | 1.979 | 3.355 | 1.922 | 2.049 | 2.194 |
| | feed_forward.w1 | 1.993 | 3.869 | 1.938 | 2.045 | 2.163 |
| | feed_forward.w2 | 1.986 | 4.162 | 1.892 | 2.094 | 2.142 |
| | feed_forward.w3 | 1.994 | 3.451 | 1.958 | 2.066 | 2.162 |
| 7 | attention.wk | 1.968 | 4.518 | 1.878 | 2.037 | 2.152 |
| | attention.wo | 1.991 | 3.691 | 1.859 | 2.005 | 2.101 |
| | attention.wq | 1.984 | 4.400 | 1.924 | 2.035 | 2.136 |
| | attention.wv | 1.978 | 3.362 | 1.920 | 2.044 | 2.188 |
| | feed_forward.w1 | 1.987 | 3.779 | 1.931 | 2.038 | 2.160 |
| | feed_forward.w2 | 1.992 | 3.977 | 1.886 | 2.138 | 2.140 |
| | feed_forward.w3 | 1.992 | 3.383 | 1.954 | 2.064 | 2.159 |

*Table 6.* Per-matrix entropy statistics and compression rates (bits/parameter) for Zstandard and LZMA on `int8`-packed weights. Calibration at CTX=2048.

**Ablation of individual components.** We now present an ablation study illustrating the contribution of each technique described in Section 4. All plots report the relative MSE at the *input* to each layer's weight matrix, comparing two quantized models against a shared unquantized reference. Filled markers denote the first configuration (run A), and hollow markers denote the second (run B).

*Residual stream correction.* Figure 6 compares WaterSIC with and without residual stream compensation (18) on Llama-3.2-1B at 4 bit. The improvement is concentrated at the down-projection layers ($w_o$ and $w_2$), as expected: these are the only layers whose objective changes under residual compensation, since they are the only ones that contribute directly to the residual stream.

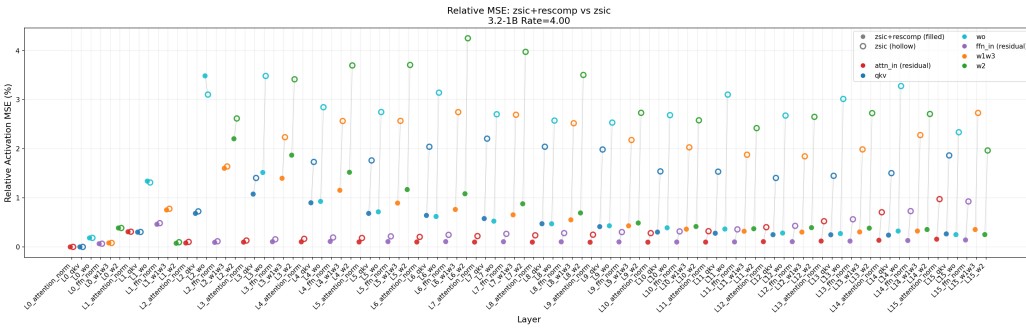

*Figure 6.* Effect of residual stream compensation on Llama-3.2-1B (4 bit). Residual compensation reduces activation MSE primarily at $w_o$ and $w_2$ inputs, with improvements propagating to subsequent layers through the residual stream. Computed at CTX=2048.

*Activation drift correction.* Figure 7 adds activation drift correction (Qronos) [17] on top of residual compensation. The drift-corrected statistics improve most layers, but notably *increase* the relative MSE at $w_o$ inputs in several layers. This occurs because the softmax nonlinearity amplifies small errors in the QKV projections: when the drift-corrected Hessian is slightly misspecified, the resulting quantization errors in $w_q, w_k, w_v$ are magnified through attention, producing worse $w_o$ input distortion than using the standard Hessian.

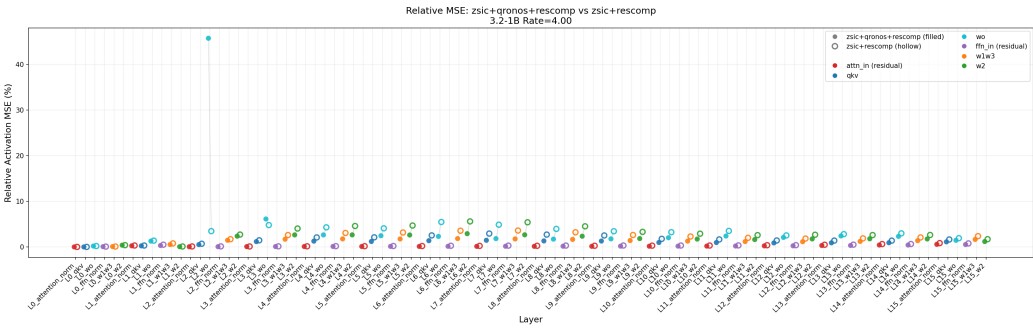

*Figure 7.* Effect of activation drift correction on Llama-3.2-1B (4 bit). Adding Qronos improves most layers but degrades $w_o$ inputs in some layers, where softmax amplifies QKV quantization errors. Computed at CTX=2048.

*Attention-weighted calibration.* Figure 8 shows that adding attention-weighted calibration [19] with joint adaptive mixing [58] resolves the $w_o$ degradation introduced by Qronos. By reweighting the QKV covariance estimates to account for the attention sink at position 0 and jointly optimizing the mixing parameters to minimize $w_o$ input MSE [60], the quantizer produces QKV weights whose errors are no longer amplified through softmax.

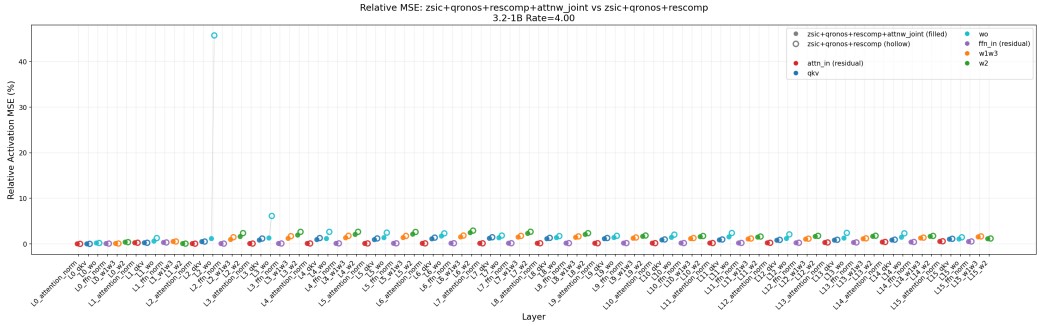

*Figure 8.* Effect of attention-weighted calibration on Llama-3.2-1B (4 bit). Joint adaptive mixing with attention-weighted covariances eliminates the $w_o$ degradation caused by Qronos. Computed at CTX=2048.

*Adaptive mixing for larger models.* While the combination of Qronos, residual compensation, and attention weighting works well on smaller models, we found that activation drift correction becomes increasingly unstable in deeper layers of larger

models, where $\hat{X}$ drifts further from $X$. Figure 9 demonstrates this on Qwen3-8B (36 layers): the adaptive mixing procedure from (58) stabilizes Qronos by interpolating toward the original statistics when the drift-corrected Hessian is ill-conditioned, yielding consistent improvements across layers.

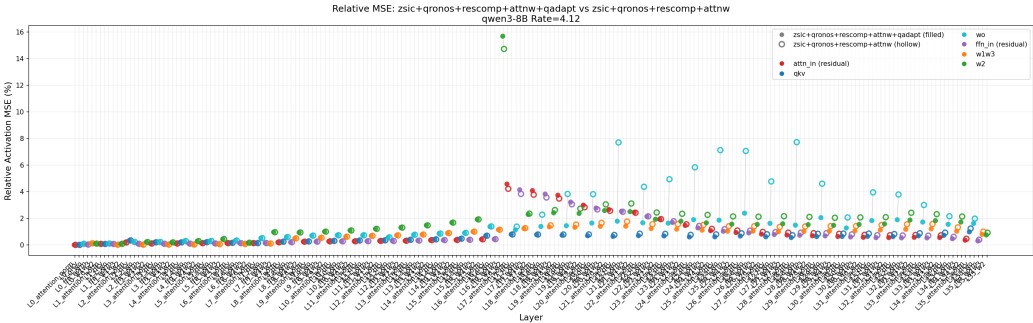

*Figure 9.* Effect of adaptive mixing on Qwen3-8B (4.12 bit). Adaptive $\epsilon_{qr}$ blending stabilizes Qronos in deeper layers where pure drift correction degrades. Computed at CTX=2048.

*Cumulative effect.* Finally, Figure 10 compares the full pipeline (WaterSIC with residual compensation, Qronos, attention weighting, and adaptive mixing) against base WaterSIC on Llama-3.2-1B. The cumulative improvement is substantial and consistent across all layer types, with the largest gains at $w_o$ and $w_2$ inputs, where the individual contributions compound.

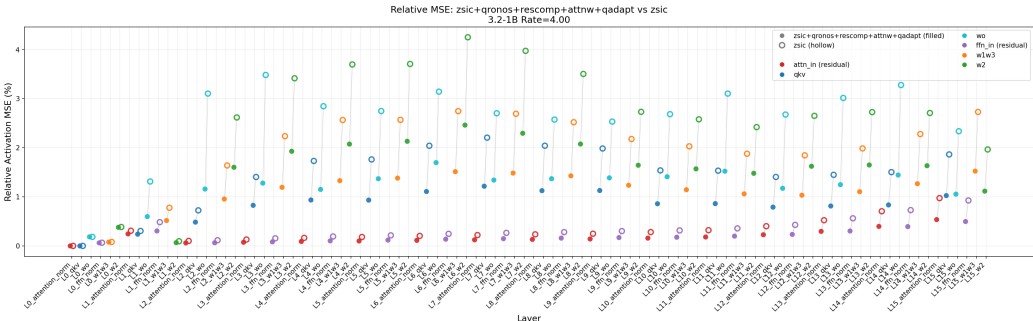

*Figure 10.* Full WaterSIC pipeline vs. base WaterSIC on Llama-3.2-1B (4 bit). All techniques combined yield a consistent reduction in activation MSE across every layer. Computed at CTX=2048.

**Gaussianity of weights.** Our theoretical analysis assumes i.i.d. Gaussian rows of $W$. We expect this to be representative of modern LLMs for two reasons. First, the random Hadamard transform applied during incoherence processing Gaussianizes weight outliers, since the resulting entries are linear combinations of $\Theta(n)$ original entries weighted by random signs. Second, the outlier features common in early LLMs have largely disappeared in models trained with modern optimizers, layer-norm placements, and learning rate schedules (Park et al., 2025). To verify this empirically, we compute, for each weight matrix in Llama-2-7B, the Kolmogorov–Smirnov distance between the empirical CDF and its best-fit Gaussian and Laplace CDFs, and report the per-layer-type averages over all 32 decoder layers in Figure 11.

The FFN matrices $(w_1, w_2, w_3)$ together with the attention value and output projections $(w_v, w_o)$ all admit a near-perfect Gaussian fit, with $D_{\mathrm{Gauss}} < 1\%$ in every layer and the Gaussian preferred over Laplace in $\geq 30/32$ layers per type. The query and key projections $(w_q, w_k)$ deviate the most: their tails are slightly heavier, the Gaussian and Laplace fits are within $\sim 0.3\%$ of each other on average, and the Laplace fit is preferred in roughly half of the layers for $w_k$. We observe the same pattern on Llama-3-8B and Llama-3.2-1B: the FFN matrices are consistently Gaussian, and $w_q, w_k$ are the only matrices for which a Laplace fit is occasionally closer. Figure 11 shows representative weight histograms with best-fit Gaussian and Laplace overlays for several layers of Llama-3.2-1B.

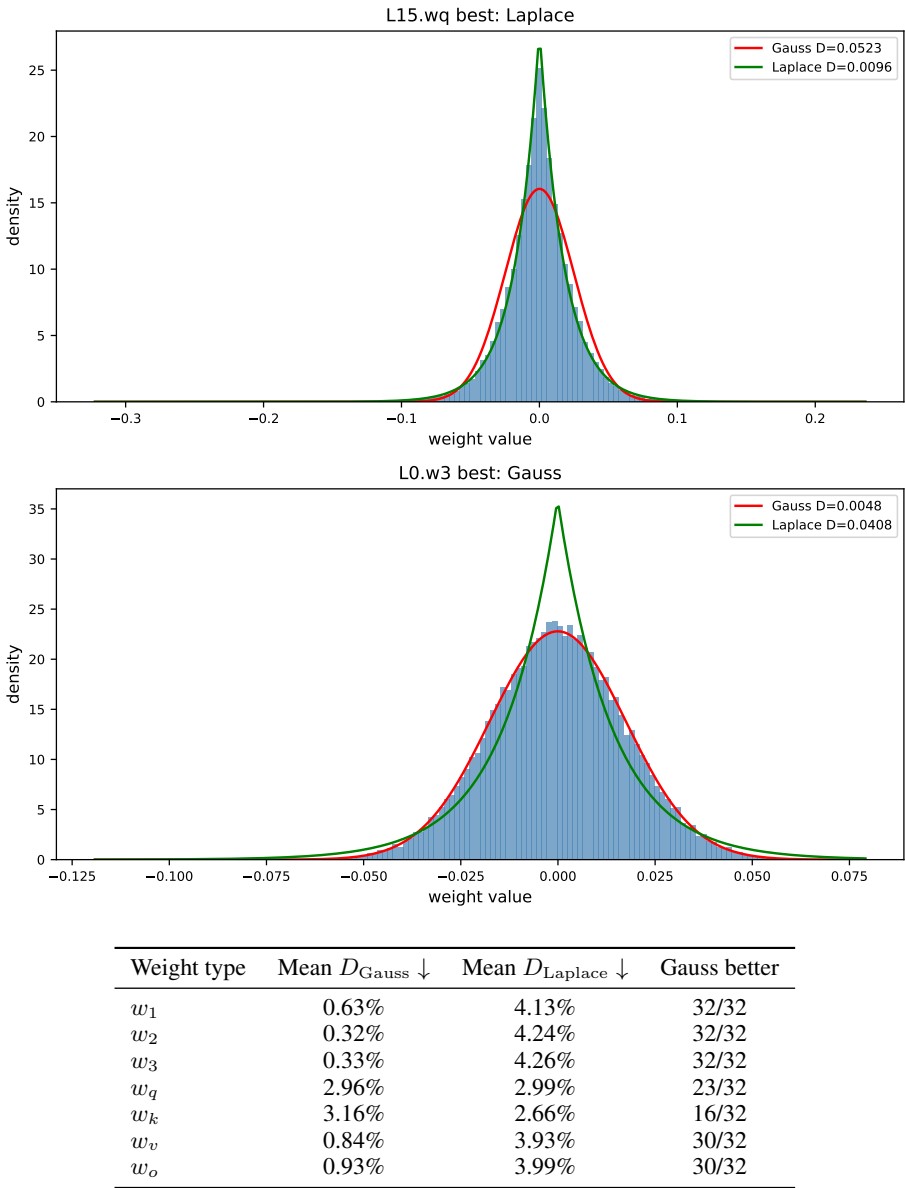

| Weight type | Mean $D_{\text{Gauss}} \downarrow$ | Mean $D_{\text{Laplace}} \downarrow$ | Gauss better |
|---|---|---|---|
| $w_1$ | 0.63% | 4.13% | 32/32 |
| $w_2$ | 0.32% | 4.24% | 32/32 |
| $w_3$ | 0.33% | 4.26% | 32/32 |
| $w_q$ | 2.96% | 2.99% | 23/32 |
| $w_k$ | 3.16% | 2.66% | 16/32 |
| $w_v$ | 0.84% | 3.93% | 30/32 |
| $w_o$ | 0.93% | 3.99% | 30/32 |

*Figure 11.* The figure contains weight histograms for selected layers of Llama-3.2-1B with best-fit Gaussian and Laplace overlays. The table contains the Kolmogorov-Smirnov distance to the best-fit Gaussian and Laplace distributions averaged over the 32 decoder layers of Llama-2-7B. The rightmost column reports the number of layers (out of 32) for which the Gaussian fit is closer than the Laplace fit.

## F. Results on additional evals and additional models.

**Llama-3.2-1B.** We report rate-distortion results on Llama-3.2-1B in Table 7. WaterSIC is calibrated on WikiText-2 train at CTX=2048, and WaterSIC-FT denotes the same model after post-quantization finetuning on WikiText-2 at CTX=2048; both are evaluated at CTX=2048 on WikiText-2 test and C4 validation. Because calibration is performed exclusively on WikiText-2, the C4 gap to BF16 is wider than the W2 gap at low rates. For instance, at 1 bit WaterSIC-FT reaches W2 PPL 32.85 (23.12 above the BF16 reference of 9.73) but C4 PPL 160.70 (147.93 above the BF16 reference of 12.77), and finetuning on WikiText-2 alone does not close this gap. We examine the role of the calibration and finetuning corpus in detail in the next section.

*Table 7.* Llama-3.2-1B WikiText-2 and C4 test perplexity at various rates. WaterSIC is calibrated on WikiText-2 train at CTX=2048; WaterSIC-FT is the same model after post-quantization finetuning on WikiText-2 at CTX=2048. Both evaluated at CTX=2048. Unquantized (BF16): W2 PPL 9.73, C4 PPL 12.77.

| Rate | WaterSIC W2↓ | WaterSIC C4↓ | WaterSIC-FT W2↓ | WaterSIC-FT C4↓ |
|------|--------------|--------------|------------------|------------------|
| 1.00 | 82.45 | 643.29 | 32.85 | 160.70 |
| 1.50 | 30.73 | 140.72 | 19.16 | 59.97 |
| 2.00 | 16.19 | 37.73 | 13.59 | 27.80 |
| 2.50 | 11.83 | 18.81 | 11.34 | 17.80 |
| 3.00 | 10.57 | 15.26 | 10.45 | 15.06 |
| 3.18 | 10.35 | 14.73 | 10.27 | 14.61 |
| 3.50 | 10.11 | 14.03 | 10.08 | 13.99 |
| 3.76 | 9.99 | 13.62 | 9.98 | 13.60 |
| 4.00 | 9.92 | 13.35 | 9.91 | 13.34 |

To complement the perplexity analysis, we also report the KL divergence between the unquantized (BF16) and quantized model output distributions, evaluated on WikiText-2 test:

$$\text{KLD} \;=\; \text{KL}(P_{\text{BF16}} \,\|\, P_{\text{quant}}) \;=\; \sum_x P_{\text{BF16}}(x) \, \log \frac{P_{\text{BF16}}(x)}{P_{\text{quant}}(x)}.$$

Figure 12 plots this quantity against the average bitwidth for Huffman-GPTQ, WaterSIC, and WaterSIC-FT; the shaded region marks the KL gap between Huffman-GPTQ and WaterSIC at matched bitwidth. WaterSIC significantly reduces the KL divergence over Huffman-GPTQ, especially at low bitrates, and post-quantization finetuning further reduces it.

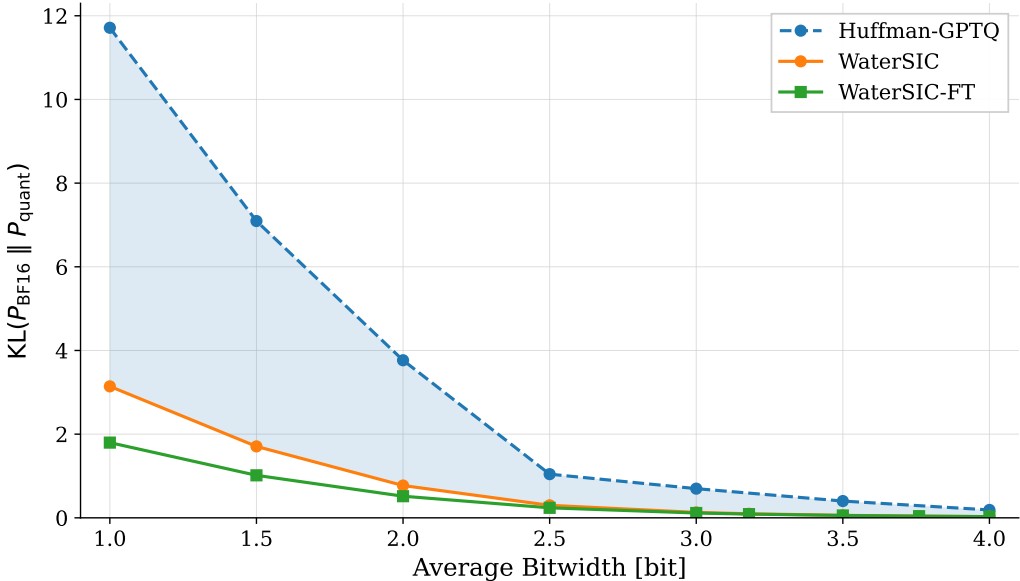

*Figure 12.* Llama-3.2-1B: KL divergence between BF16 and quantized output distributions, $\text{KL}(P_{\text{BF16}} \,\|\, P_{\text{quant}})$, against average bitwidth, for Huffman-GPTQ, WaterSIC, and WaterSIC-FT. The shaded region marks the gap between Huffman-GPTQ and WaterSIC at matched bitwidth. Evaluated at CTX=2048.

**Qwen3-8B.** We report rate-distortion results on Qwen3-8B in Table 8. The calibration, finetuning, and evaluation protocol matches Llama-3.2-1B above: WaterSIC is calibrated on WikiText-2 train at CTX=2048, WaterSIC-FT is the same model after post-quantization finetuning on WikiText-2 at CTX=2048, and both are evaluated at CTX=2048 on WikiText-2 test and C4 validation.

*Table 8.* Qwen3-8B WikiText-2 and C4 test perplexity at various rates. WaterSIC is calibrated on WikiText-2 train at CTX=2048; WaterSIC-FT is the same model after post-quantization finetuning on WikiText-2 at CTX=2048. Both evaluated at CTX=2048. Unquantized (BF16): W2 PPL 9.73, C4 PPL 13.30.

| Rate | WaterSIC W2↓ | WaterSIC C4↓ | WaterSIC-FT W2↓ | WaterSIC-FT C4↓ |
|------|------|------|------|------|
| 1.12 | 25.59 | 102.52 | 14.54 | 43.17 |
| 1.62 | 14.45 | 33.96 | 11.27 | 24.33 |
| 2.12 | 11.37 | 18.64 | 10.18 | 16.54 |
| 2.62 | 10.44 | 15.15 | 9.85 | 14.50 |
| 3.12 | 10.03 | 14.11 | 9.73 | 13.83 |
| 3.62 | 9.87 | 13.77 | 9.70 | 13.63 |
| 4.12 | 9.79 | 13.57 | 9.67 | 13.49 |

**Llama-3-8B.** We report rate-distortion results on Llama-3-8B in Table 9 and Figure 13. The protocol matches Llama-3.2-1B and Qwen3-8B above: calibration on WikiText-2 train at CTX=2048, evaluation on WikiText-2 test at CTX=2048, and (for WaterSIC-FT) finetuning on WikiText-2 train at CTX=2048; the unquantized (BF16) reference is W2 PPL 6.14.

*Table 9.* Comparison of WikiText-2 perplexity results on Llama-3-8B, evaluated at CTX=2048. WaterSIC-FT is finetuned on WikiText-2 at CTX=2048. In each rate group, WaterSIC-FT achieves the best PPL at minimal rate. The unquantized (BF16) model perplexity is 6.14.

| Method | Avg. Bitwidth | WikiText-2 PPL |
|--------|------|------|
| WaterSIC-FT | **1.00** | **18.97** |
| WaterSIC | 1.00 | 47.15 |
| WaterSIC-FT | **1.50** | **10.90** |
| WaterSIC | 1.50 | 15.89 |
| Huffman-GPTQ | 1.91 | 22.79 |
| WaterSIC-FT | **2.00** | **8.06** |
| WaterSIC | 2.00 | 8.93 |
| GPTVQ | 2.12 | 9.94 |
| GPTVQ | 2.25 | 9.59 |
| WaterSIC-FT | **2.50** | **7.04** |
| WaterSIC | 2.50 | 7.25 |
| Huffman-GPTQ | 2.51 | 11.46 |
| Huffman-GPTQ | 2.94 | 8.57 |
| WaterSIC-FT | **3.00** | **6.60** |
| WaterSIC | 3.00 | 6.65 |
| GPTVQ | 3.12 | 7.00 |
| WaterSIC-FT | **3.18** | **6.50** |
| WaterSIC | 3.18 | 6.53 |
| NestQuant (W-only) | 3.18 | 6.70 |
| Huffman-GPTQ | 3.41 | 7.36 |
| WaterSIC-FT | **3.50** | **6.37** |
| WaterSIC | 3.50 | 6.38 |
| NestQuant (W-only) | 3.50 | 6.49 |
| WaterSIC-FT | **3.76** | **6.30** |
| WaterSIC | 3.76 | 6.31 |
| NestQuant (W-only) | 3.76 | 6.38 |
| WaterSIC-FT | **4.00** | **6.25** |
| WaterSIC | 4.00 | 6.26 |
| NestQuant (W-only) | 3.99 | 6.31 |
| AWQ (W4A16 g128) | 4.00 | 6.54 |
| Huffman-GPTQ | 3.97 | 6.74 |

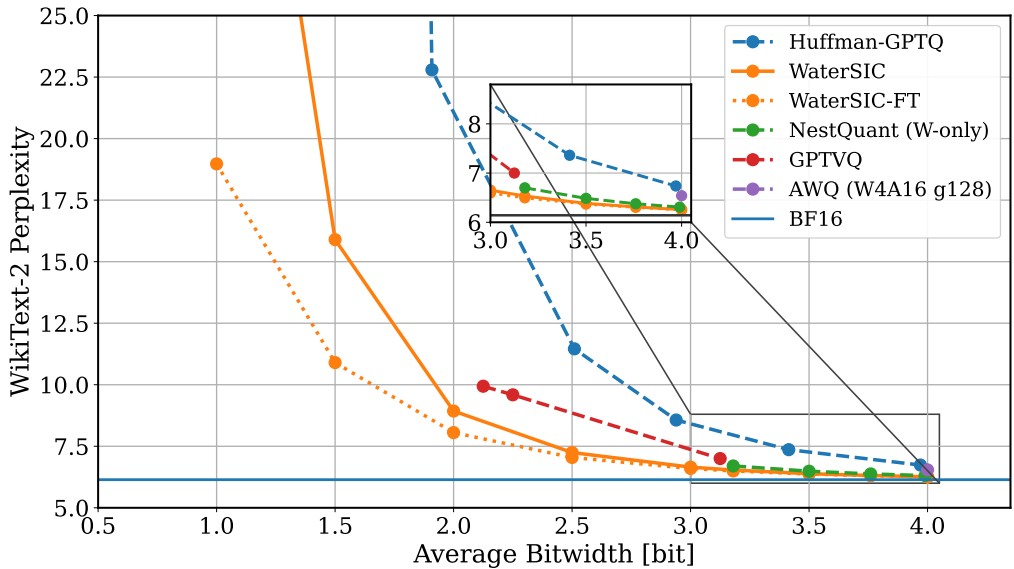

*Figure 13.* Llama-3-8B: WaterSIC vs. other algorithms. WaterSIC and Huffman-GPTQ report rates as entropy; the other methods use log-cardinality. Evaluated at CTX=2048.

To compare against prior 2-bit methods we additionally re-evaluate WaterSIC at the longer context length 8192 used in those works, with finetuning likewise performed at CTX=8192. Table 10 reports the comparison; baseline numbers are taken from (Malinovskii et al., 2024).

*Table 10.* Llama-3-8B 2-bit comparison, evaluated at context length 8192. WaterSIC is calibrated on WikiText-2 train at CTX=2048 and finetuned at CTX=8192. Baseline numbers from (Malinovskii et al., 2024).

| Method | Bitwidth | W2 PPL↓ | C4 PPL↓ |
|---|---|---|---|
| BF16 | 16 | 5.54 | 7.10 |
| QuIP | 2.01 | 76.95 | 98.47 |
| DB-LLM | 2.01 | 12.77 | 14.82 |
| PV-Tuning | 2.01 | **6.99** | **8.29** |
| WaterSIC (no FT) | 2.00 | 8.01 | 13.48 |
| WaterSIC (FT W2) | 2.00 | 7.20 | 11.69 |
| WaterSIC (FT C4) | 2.00 | 7.39 | 9.61 |
| WaterSIC (FT RP) | 2.00 | 7.35 | 9.60 |

**Llama-2-7B.** We report rate-distortion results on Llama-2-7B in Table 11 and Figure 14, again under the calibration / finetuning / evaluation protocol of the previous models (CTX=2048 throughout); the unquantized (BF16) reference is W2 PPL 5.47.

*Table 11.* Comparison of WikiText-2 perplexity results on Llama-2-7B, evaluated at CTX=2048. WaterSIC-FT is finetuned on WikiText-2 at CTX=2048. In each rate group, WaterSIC-FT achieves the best PPL at minimal rate. The unquantized (BF16) model perplexity is 5.47.

| Method | Avg. Bitwidth | WikiText-2 PPL |
|---|---|---|
| WaterSIC-FT | **1.00** | **12.03** |
| WaterSIC | 1.00 | 22.43 |
| WaterSIC-FT | **1.50** | **7.64** |
| WaterSIC | 1.50 | 8.97 |
| Huffman-GPTQ | 1.53 | 83.23 |
| Huffman-GPTQ | 1.95 | 55.84 |
| WaterSIC-FT | **2.00** | **6.29** |
| WaterSIC | 2.00 | 6.57 |
| QuIP# | 2.00 | 6.66 |
| GPTVQ | 2.12 | 7.18 |
| GPTVQ | 2.25 | 6.99 |
| WaterSIC-FT | **2.50** | **5.81** |
| WaterSIC | 2.50 | 5.88 |
| Huffman-GPTQ | 2.55 | 6.38 |
| WaterSIC-FT | **3.00** | **5.63** |
| WaterSIC | 3.00 | 5.65 |
| QuIP# | 3.00 | 5.79 |
| GPTVQ | 3.12 | 5.83 |
| Huffman-GPTQ | 2.98 | 5.96 |
| WaterSIC-FT | **3.50** | **5.54** |
| WaterSIC | 3.50 | 5.55 |
| Huffman-GPTQ | 3.46 | 5.74 |
| WaterSIC-FT | **4.00** | **5.50** |
| WaterSIC | 4.00 | 5.51 |
| NestQuant (W-only) | 4.00 | 5.53 |
| QuIP# | 4.00 | 5.56 |
| AWQ (W4A16 g128) | 4.00 | 5.60 |
| Huffman-GPTQ | 4.01 | 5.62 |

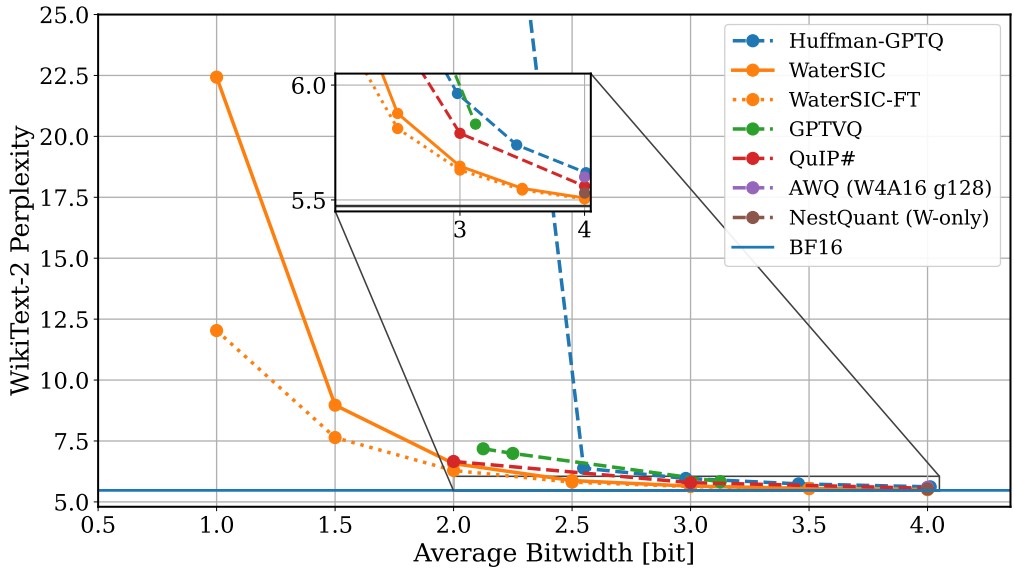

*Figure 14.* Llama-2-7B: WaterSIC vs. other algorithms. WaterSIC and Huffman-GPTQ report rates as entropy; the other methods use log-cardinality. Evaluated at CTX=2048.

We additionally study the sensitivity of the algorithm to calibration context length at 2 bits, this time at the longer evaluation context CTX=4096 used in prior works. In every configuration of this experiment, finetuning is performed at the evaluation context length 4096 on the full WikiText-2 training split, so the experiment isolates the effect of the calibration context. Segmenting WikiText-2 train at non-overlapping intervals yields roughly twice as many sequences at CTX=1024 (∼2378) as at CTX=2048 (∼1189), so the shorter calibration context trades capturing of longer-range dependencies for a better-

conditioned Hessian estimate. Table 12 reports WaterSIC at calibration contexts 1024 and 2048 across four finetuning configurations. The two calibration contexts produce nearly identical perplexities (within 0.03 on W2 and 0.05 on C4), and the two effects above appear to roughly cancel out, suggesting that the calibration context length has little effect on final quality once the model is finetuned at the evaluation context length.

*Table 12.* Llama-2-7B at 2 bits: effect of calibration context length on WaterSIC. Calibration on WikiText-2 train; finetuning on the full WikiText-2 training split segmented at the evaluation context length 4096. Evaluated at CTX=4096. Unquantized (BF16): W2 PPL 5.12, C4 PPL 6.63.

| Finetuning set | Calib. CTX=1024 | | Calib. CTX=2048 | |
|---|---|---|---|---|
| | W2 PPL↓ | C4 PPL↓ | W2 PPL↓ | C4 PPL↓ |
| None | 6.152 | 9.853 | 6.123 | 9.813 |
| WikiText-2 | 5.864 | 9.237 | 5.864 | 9.285 |
| C4 | 5.994 | 8.737 | 5.977 | 8.729 |
| RedPajama | 5.986 | 8.832 | 5.976 | 8.818 |

Table 13 compares WaterSIC at 2 bits and CTX=4096 against AQLM, QuIP#, DB-LLM, PV-Tuning, and YAQA; baseline numbers are taken from (Malinovskii et al., 2024) and (Tseng et al., 2025). WaterSIC's W2 PPL matches PV-Tuning to within 0.02 (5.86 vs. 5.84 with W2 finetuning), while C4 PPL trails behind, consistent with calibrating exclusively on WikiText-2. Switching the finetuning set to C4 or RedPajama recovers most of this C4 gap: against the BF16 C4 PPL of 6.63, the gap shrinks from 2.66 ($9.29 - 6.63$, with WikiText-2 finetuning) to 2.10 ($8.73 - 6.63$, with C4 finetuning).

*Table 13.* Llama-2-7B 2-bit comparison, evaluated at context length 4096. WaterSIC is calibrated on WikiText-2 train at CTX=2048 and finetuned at CTX=4096. Baseline numbers from (Malinovskii et al., 2024) and (Tseng et al., 2025).

| Method | Bitwidth | W2 PPL↓ | C4 PPL↓ |
|---|---|---|---|
| BF16 | 16 | 5.12 | 6.63 |
| AQLM | 2.02 | 6.64 | 8.56 |
| QuIP# | 2.01 | 6.19 | 8.16 |
| DB-LLM | 2.01 | 7.23 | 9.62 |
| YAQA-B (INT2) | 2.00 | 7.45 | 9.22 |
| PV-Tuning | 2.02 | **5.84** | **7.62** |
| WaterSIC (no FT) | 2.00 | 6.12 | 9.81 |
| WaterSIC (FT W2) | 2.00 | 5.86 | 9.29 |
| WaterSIC (FT C4) | 2.00 | 5.98 | 8.73 |
| WaterSIC (FT RP) | 2.00 | 5.98 | 8.82 |

**Llama-3-70B.** We report results on Llama-3-70B at 2 and 4 bits in Table 14. We ran the algorithm on 4 H100 GPUs, evaluating WikiText-2 perplexity at context length 2048. At 4 bits, WaterSIC-FT achieves the lowest perplexity of 3.03. At 2 bits, WaterSIC-FT reaches 5.68, substantially outperforming all scalar PTQ baselines for which comparable results have been reported.

*Table 14.* Llama-3-70B WikiText-2 perplexity (↓) at CTX=2048 under 2-bit and 4-bit weight-only quantization. The unquantized (BF16) model perplexity is 2.86. RTN, QuIP, GPTQ, AWQ, and PB-LLM results are from Huang et al. (2024).

| Method | 2 bits | 4 bits |
|---|---|---|
| RTN | $4.6 \times 10^5$ | 3.60 |
| QuIP | 13.0 | 3.40 |
| GPTQ | 11.9 | 3.30 |
| AWQ | $1.7 \times 10^6$ | 3.30 |
| PB-LLM | 11.6 | – |
| OstQuant | – | 3.19 |
| NestQuant | – | 3.14 |
| WaterSIC | 6.96 | 3.04 |
| WaterSIC-FT | **5.68** | **3.03** |

**Effect of Calibration and Finetuning Set.** We investigate how the choice of calibration and finetuning datasets affects

downstream perplexity, focusing on Llama-3.2-1B at 2 bits. We vary the calibration set between the WikiText-2 train split and a 1T-token sample of RedPajama, using the same number of sequences in both (1189, equal to the WikiText-2 train split segmented at CTX=2048). We additionally vary the finetuning set across four options: no finetuning, WikiText-2, RedPajama, and C4, using 1189 sequences of length 2048 in each case. We evaluate both WikiText-2 test PPL and C4 test PPL at CTX=2048.

*Table 15.* Llama-3.2-1B at 2 bits: effect of the calibration and finetuning sets on WaterSIC. Calibration on WikiText-2 train or RedPajama at CTX=2048; finetuning at CTX=2048 on the indicated set. Evaluated at CTX=2048. Unquantized (BF16): W2 PPL 9.73, C4 PPL 12.77.

| Finetuning set | Calibration on WikiText-2 | | Calibration on RedPajama | |
|---|---|---|---|---|
| | W2 PPL↓ | C4 PPL↓ | W2 PPL↓ | C4 PPL↓ |
| None | 16.19 | 37.73 | 27.43 | 30.33 |
| WikiText-2 | 13.59 | 27.80 | 12.51 | 17.73 |
| RedPajama | 14.52 | 23.78 | 14.52 | 17.40 |
| C4 | 14.56 | 23.53 | 14.76 | 17.37 |

Finetuning improves both W2 and C4 PPL regardless of the calibration set. Without finetuning, calibrating on WikiText-2 gives lower W2 PPL but higher C4 PPL than calibrating on RedPajama (W2/C4 of 16.19/37.73 vs. 27.43/30.33), reflecting the calibration–evaluation distribution mismatch. Surprisingly, the lowest W2 PPL of any configuration (12.51) is reached by calibrating on RedPajama and then finetuning on WikiText-2, which beats the matched WikiText-2-calibrated, WikiText-2-finetuned setting (13.59); this suggests that the broader RedPajama calibration produces statistics that finetuning can exploit more effectively. C4 PPL, on the other hand, is minimized by RedPajama or C4 finetuning regardless of the calibration set (17.37–17.73 when calibrated on RedPajama; 23.53–23.78 when calibrated on WikiText-2).

We extend this analysis across rates from 1 to 4 bits for the WikiText-2-calibrated Llama-3.2-1B model. Figure 15 shows rate-distortion curves for no finetuning, finetuning on WikiText-2, and finetuning on RedPajama, evaluated on WikiText-2 test PPL (left) and C4 test PPL (right). Table 16 reports the full numerical results.

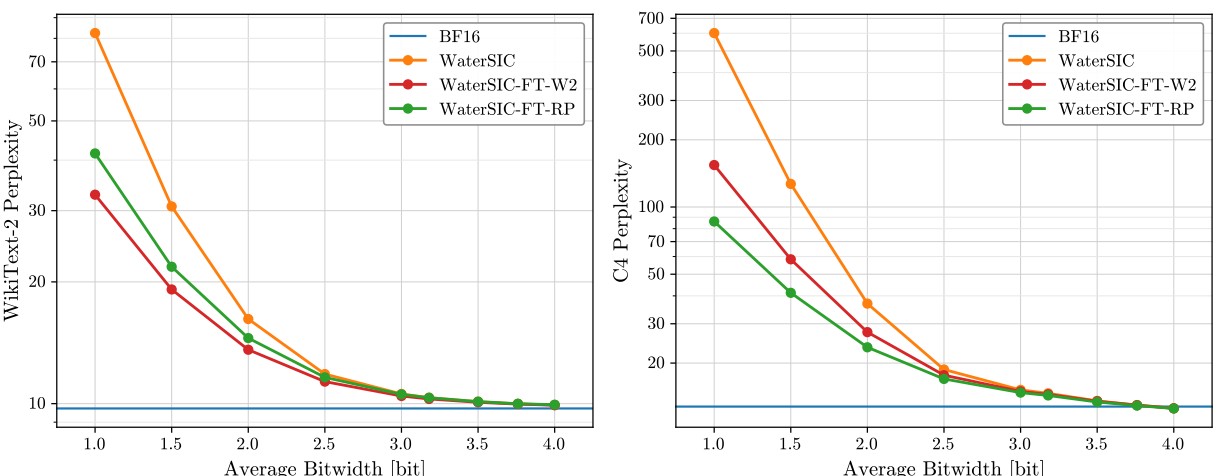

*Figure 15.* Rate-distortion curves for Llama-3.2-1B calibrated on WikiText-2, evaluated on WikiText-2 test PPL (left) and C4 test PPL (right) for three configurations: no finetuning, finetuning on WikiText-2, and finetuning on RedPajama. Evaluated at CTX=2048.

*Table 16.* Llama-3.2-1B WikiText-2 and C4 test perplexity at various rates. Calibrated on WikiText-2 train at CTX=2048; finetuning at CTX=2048. Unquantized (BF16): W2 PPL 9.73, C4 PPL 12.77.

| Rate | W2 PPL↓ | | | C4 PPL↓ | | |
|------|---------|--------|-------|---------|--------|--------|
|      | No FT   | FT W2  | FT RP | No FT   | FT W2  | FT RP  |
| 1.00 | 82.45   | 32.85  | 41.55 | 643.29  | 160.70 | 88.26  |
| 1.50 | 30.73   | 19.16  | 21.80 | 140.72  | 59.97  | 41.84  |
| 2.00 | 16.19   | 13.59  | 14.52 | 37.73   | 27.80  | 23.78  |
| 2.50 | 11.83   | 11.34  | 11.62 | 18.81   | 17.80  | 17.16  |
| 3.00 | 10.57   | 10.45  | 10.54 | 15.26   | 15.06  | 14.91  |
| 3.18 | 10.35   | 10.27  | 10.34 | 14.73   | 14.61  | 14.49  |
| 3.50 | 10.11   | 10.08  | 10.12 | 14.03   | 13.99  | 13.89  |
| 3.76 | 9.99    | 9.98   | 9.99  | 13.62   | 13.60  | 13.54  |
| 4.00 | 9.92    | 9.91   | 9.94  | 13.35   | 13.34  | 13.31  |

The benefit of finetuning is concentrated at low rates (1–2.5 bits). For example, at 1 bit RedPajama finetuning takes C4 PPL from 643.29 down to 88.26 (an 86% reduction) and WikiText-2 finetuning takes W2 PPL from 82.45 down to 32.85 (a 60% reduction). At rates $\geq 3.5$ bits the differences between finetuning configurations shrink to within 0.15 PPL on either evaluation, since the quantization error is by then small relative to the model's intrinsic approximation error. Comparing the two finetuning sets at low rates, WikiText-2 finetuning gives lower W2 PPL (32.85 vs. 41.55 at 1 bit) but higher C4 PPL than RedPajama finetuning (160.70 vs. 88.26 at 1 bit): the broader mixed corpus transfers better to C4, and the W2 PPL it gives up is small in absolute terms compared to the C4 PPL it recovers. This is the same domain-matching effect visible in Table 15: each finetuning set is best on its own evaluation distribution, and a broader finetuning corpus generalizes better off-distribution.

## G. Zero-shot accuracy evaluations

The previous sections focused on perplexity on WikiText-2 and C4. We now turn to downstream zero-shot accuracy across a range of standard benchmarks, to verify that the perplexity advantages of WaterSIC translate into corresponding gains on tasks the quantized model is actually deployed for.

Table 17 compares WaterSIC and HPTQ on Llama-3.2-1B across bitrates. At every rate, WaterSIC outperforms HPTQ on the majority of benchmarks and remains competitive on the rest. Table 18 compares WaterSIC against the Qwen3-8B numbers reported in (Chen et al., 2026); the same pattern holds, with WaterSIC matching or improving over the reported HPTQ numbers on most benchmarks.

| Rate | Method | ARC-C | ARC-E | HellaSwag | OBQA | PIQA | SocialIQA | Wino |
|------|--------|-------|-------|-----------|------|------|-----------|------|
| 1.50 | Huffman-GPTQ | **0.2722** | 0.2597 | 0.2619 | **0.2960** | 0.5185 | 0.3306 | 0.5067 |
|      | WaterSIC | 0.2415 | 0.3620 | 0.3127 | 0.2720 | 0.5729 | 0.3501 | 0.5107 |
|      | WaterSIC-FT | 0.2577 | **0.3653** | **0.3368** | 0.2680 | **0.5778** | **0.3547** | **0.5328** |
| 2.00 | Huffman-GPTQ | 0.2218 | 0.2959 | 0.3030 | 0.2520 | 0.5332 | 0.3465 | 0.5075 |
|      | WaterSIC | 0.2679 | 0.4655 | 0.4052 | 0.2920 | 0.6066 | 0.3685 | 0.5328 |
|      | WaterSIC-FT | **0.2910** | **0.5143** | **0.4410** | **0.3080** | **0.6333** | **0.3823** | **0.5399** |
| 2.50 | Huffman-GPTQ | 0.2534 | 0.4327 | 0.4215 | 0.2880 | 0.6132 | 0.3593 | 0.5454 |
|      | WaterSIC | 0.3166 | 0.5551 | 0.5205 | 0.3400 | 0.6774 | 0.3992 | 0.5912 |
|      | WaterSIC-FT | **0.3276** | **0.5589** | **0.5456** | **0.3520** | **0.7013** | **0.4007** | **0.5951** |
| 3.00 | Huffman-GPTQ | 0.3038 | 0.5215 | 0.5125 | 0.2960 | 0.6708 | 0.4028 | 0.5706 |
|      | WaterSIC | **0.3387** | **0.5720** | 0.5872 | 0.3580 | 0.7182 | **0.4248** | 0.5991 |
|      | WaterSIC-FT | 0.3370 | 0.5711 | **0.5999** | **0.3640** | **0.7247** | 0.4191 | **0.5998** |
| 3.50 | Huffman-GPTQ | 0.3217 | 0.5606 | 0.5613 | 0.3320 | 0.7133 | 0.4161 | 0.5872 |
|      | WaterSIC | 0.3601 | **0.6162** | 0.6166 | **0.3700** | **0.7421** | **0.4217** | 0.6140 |
|      | WaterSIC-FT | **0.3618** | 0.6124 | **0.6237** | 0.3680 | **0.7421** | 0.4181 | **0.6196** |
| 4.00 | Huffman-GPTQ | 0.3447 | 0.6040 | 0.6098 | 0.3600 | 0.7291 | 0.4212 | 0.6054 |
|      | WaterSIC | **0.3737** | 0.6170 | 0.6287 | **0.3800** | **0.7497** | 0.4340 | 0.6069 |
|      | WaterSIC-FT | 0.3686 | **0.6174** | **0.6338** | 0.3760 | 0.7492 | **0.4345** | **0.6101** |

*Table 17.* Zero-shot accuracy on Llama-3.2-1B. For each rate and task, we bold the best result among Huffman-GPTQ, WaterSIC, and WaterSIC-FT (higher is better). Quantization calibrated at CTX=2048.

| Rate | Method | Wino | MMLU | HSwag | PIQA | SciQ | TQA-MC1 | TQA-MC2 |
|---|---|---|---|---|---|---|---|---|
| 16.000 | BF16 | 68.11 | 73.02 | 74.90 | 77.80 | 95.70 | 36.35 | 54.50 |
| 4.125 | Huffman-GPTQ | 67.17 | 72.28 | **74.84** | 77.42 | 95.60 | 35.01 | 53.36 |
| | GPTQ | 68.82 | 71.76 | 74.22 | 77.58 | 95.30 | 36.35 | 54.55 |
| | HRTN | 67.56 | 72.15 | 74.72 | 76.99 | 94.20 | 36.47 | **56.46** |
| | RTN | 67.17 | 69.71 | 74.30 | 75.90 | 94.50 | **36.84** | 55.77 |
| | WaterSIC | **70.09** | **72.77** | 74.60 | 76.99 | **95.90** | 35.99 | 53.80 |
| | WaterSIC-FT | 68.75 | 72.33 | 74.35 | **77.69** | 95.40 | 35.62 | 54.12 |
| 3.125 | Huffman-GPTQ | 66.93 | **70.96** | 73.18 | **77.53** | 95.40 | 36.11 | 54.73 |
| | GPTQ | 68.35 | 65.80 | 70.80 | 75.46 | 75.46 | 36.11 | **55.21** |
| | HRTN | 66.22 | 67.85 | 72.36 | 76.12 | 93.70 | 35.13 | 53.68 |
| | RTN | 57.93 | 47.90 | 55.41 | 70.89 | 87.10 | 34.03 | 52.76 |
| | WaterSIC | **68.43** | 70.53 | **73.38** | 76.99 | 94.70 | **36.60** | 53.95 |
| | WaterSIC-FT | 67.72 | 70.39 | 73.36 | 77.04 | **95.40** | 34.64 | 53.24 |
| 2.125 | Huffman-GPTQ | 59.19 | 52.99 | 63.86 | 72.52 | 86.80 | 31.09 | 49.01 |
| | GPTQ | 52.25 | 34.25 | 39.32 | 57.83 | 57.83 | 28.40 | 46.91 |
| | HRTN | 51.22 | 33.91 | 49.27 | 65.78 | 76.80 | 30.48 | **51.78** |
| | RTN | 49.08 | 22.95 | 26.04 | 51.63 | 21.20 | 24.11 | 47.33 |
| | WaterSIC | 67.01 | 61.14 | 64.52 | **74.05** | 92.60 | 32.80 | 49.11 |
| | WaterSIC-FT | **68.90** | **61.69** | **67.02** | 73.99 | **93.70** | **33.41** | 50.84 |

*Table 18.* Zero-shot accuracy on Qwen3-8B. Bold indicates the best value within each rate block for each benchmark (higher is better). Quantization calibrated at CTX=2048.

