# OpenReview forum: "WaterSIC: Information-Theoretically (Near) Optimal \\Linear Layer Quantization"
_ICML.cc/2026/Conference — ICML 2026 spotlight_

### Official Review · Reviewer_x7Tw · 2026-03-03

**Soundness:** 2
**Presentation:** 4
**Significance:** 3
**Originality:** 4
**Overall Recommendation:** 5
**Confidence:** 3

**Summary:**

The paper proposes a new post-training quantization method (WaterSIC) to compress neural network linear layers by choosing different quantization granularities for different input-feature columns, guided by an information-theoretic perspective on how bits should be allocated. It argues that the popular quantization method GPTQ can be far from the best achievable tradeoff between model error and storage, and provides a theoretical analysis showing the proposed WaterSIC method can be close to an ideal limit under simplifying assumptions. Experiments on multiple language models show improved perplexity at the same average storage cost compared to several prior weight-only baselines.

**Compliance With Llm Reviewing Policy:**

Affirmed.

**Final Justification:**

The authors resolve most of my concerns. I agree that estimating the exact $R_{\mathrm{GPTQ}}$ and $R_{\mathrm{WaterSIC}}$ for a realistic distortion $D$ is an open problem, and making an optimized decompression GPU kernel for the proposed compression format is non-trivial. I think these do not limit the merits of this paper, and I would like to recommend acceptance. I have increased my rating (4->5).

**Key Questions For Authors:**

1. The theory models the weights as iid zero-mean Gaussian samples, while the actual weights are known to be heavy-tailed, e.g., Laplacian. Putting aside this theory-practice difference, there still seems to be a contradiction. In GPTQ, after each quantization step, the remaining unquantized weights are updated via error propagation. Quantizing the remaining weights can be viewed as a recursive sub-problem of GPTQ [1]. In this sub-problem, the weights are updated and deviated from the original distribution, so the sub-problem is not bounded by the theory. The distribution shift is accumulated as $n$ becomes larger. If the sub-problem is not bounded, how is the final output bounded by the theory? Furthermore, the weight update depends on the off-diagonal entries of the $L$ matrix, but the bounds $D\_{\mathrm{GPTQ}}^\*$ and $D\_{\mathrm{WaterSIC}}^\*$ only involve the diagonal entries of $L$. I hope the authors can clarify whether any non-asymptotic behavior depends on the off-diagonals.

2. This question is related to the first weakness. Theorem 3.3 takes the limit $D \to 0$. At $D \to 0$, all numbers should be perfectly preserved with little quantization error, and $R_{\mathrm{GPTQ}}$, $R_{\mathrm{WF}}$, and $R_{\mathrm{WaterSIC}}$ should be very high. In this case, the differences between the $R$ values would be relatively very small compared to the $R$ values themselves. What is the implication of this theorem? The methods seem to yield similar rates, and there is little benefit to using WaterSIC.

3. The term $\frac{2 \pi e}{12}$ frequently appears in the paper. Is there a reason not to simplify to $\frac{\pi e}{6}$?

---

References

[1] Johann Birnick. The Lattice Geometry of Neural Network Quantization: A Short Equivalence Proof of GPTQ and Babai's algorithm. ICLR 2026. URL https://openreview.net/forum?id=L4SwHIZEaJ

**Limitations:**

See the first and second weaknesses.

**Strengths And Weaknesses:**

**Strengths**

1. The paper offers a principled rate-distortion perspective on post-training quatization and uses it to justify uneven precision allocation across input-feature columns instead of a uniform setting. This gives the method a clear rationale beyond heuristics.

2. The proposed WaterSIC method is clearly specified: it extends the GPTQ pipeline with per-column step sizes and adds corrections (e.g., LMMSE, drift/residual effects, diagonal rescalers) that have been shown useful in quantization literature.

3. Experiments on multiple models report improved quality-rate tradeoffs on perplexity and zero-shot benchmarks compared to the state-of-the-art baselines.

---

**Weaknesses**

1. The paper's strongest empirical gains appear exactly where asymptotic approximations in the theory are least trustworthy (very low bits). Section 3.2 (e.g., Equation 11, Theorem 3.3) relies on approximations in the high-rate low-distortion settings, while the typical quantization settings (1-4 bits) are low-rate high-distortion. Figures 1 and 2 show the improvements compared to other methods are larger at lower bitwidth. It is not clear how the theory generalizes to the actual settings.

2. The authors should clarify if WaterSIC is only intended as a storage/transmission format rather than a speedup one. The resulting data format of WaterSIC is not practical for running LLMs on the current computer hardware. The quantized integers are encoded using entropy coding (e.g., Huffman encoding, LZ4, Zstd), which is not natively supported by the matrix multiplication instructions of the GPU. A decoding procedure (maybe sequential, non-parallel) is required at inference time and can significantly decrease the throughput. The authors admit in the limitation paragraph that they did not benchmark the throughput. But I would consider it a practically hard issue rather than simply "did not benchmark".

*Minor Problems*

3. I do not agree with the statement "all existing algorithms fix the same quantization rate for all columns of $W$" in the introduction section. Some prior works have already implemented mixed precision across groups/channels:

- Ashkboos et al. QUIK: Towards End-to-end 4-Bit Inference on Generative Large Language Models. EMNLP 2024. URL https://aclanthology.org/2024.emnlp-main.197/

- ExLlamaV2/EXL2. URL https://github.com/turboderp-org/exllamav2?tab=readme-ov-file#exl2-quantization

4. The equation of $D$ in Section 3.1 seems to miss a $\frac{1}{n}$ factor compared to Equations 1 and 4. The variable $\tau$ in Equation 2 is not defined in the paper. It would also be beneficial to state whether the logarithms are base 2 or base $e$, since the constants matter in the equations.

5. Citation formats mismatch. Some citations should be cited as text without parentheses. e.g., "(Tseng et al., 2025a)" on page 2.

---

> ### Author Rebuttal · Authors · 2026-03-31
>
> Thank you for a careful reading and insightful comments!
>
> Q1: This analysis is indeed the main theoretical contribution of our paper! And you question was puzzling us too. First, let us mention that analysis is fully rigorous for the case of W being iid N(0,\sigma^2) weights, so there is really no hidden assumptions such as “recursive updates do not change the distribution of entries”, they totally do. However, there are two things to notice. One is that our Theorem 3.3 is a low-distortion limit and in that limit, the noise is infinitesimally small and hence the propagated errors are negligible. This point is seen in Lemma B.1 eq. (26): as alpha->0 the first term is giant, whereas the “feedback noise” is bounded by a constant, which does depend on off-diagonal entries. So the answer is that if we could get R_{GPTQ}(D) for finite D (an important open problem), we would indeed see that it has some dependence on off-diagonal entries of L, which however (in our experiments) washes out already at rates > 3 bits and the asymptotic “high-rate” theory kicks in. Second thing to point out is that the Y vector on which we are performing feedback (this is crucial: unlike other analyses of GPTQ, we are thinking in the “lattice space”) is itself produced with the help of L: Y=WL. This allows us to cancel the L^{-1} term in (31) multiplying Y to get L. Ie large feedback appear only when the vector Y itself has large entries (which appear from multiplying small W by large L).
>
>
> Q2: The R_GPTQ, R_WaterSIC are both unfortunately unknown and possibly very complicated functions of the Hessian, distortion and per-channel scales (for WaterSIC). How does one understand and compare such functions? How do we optimize per-channel scales without recoursing to myriad of simulations? We proposed to look at the limit of D->0 and found that while all rates explode to infinity in this limit, they maintain a constant (and different!) gap to the fundamental lower bound R_WF (that is Thm 3.3). Looking to minimize that asymptotic gap to R_WF suggested to us how to choose per-channel scales. We then made a leap of faith (common in natural sciences, esp. phenomenology) that these same scales will continue to work well for practically relevant distortions D>0. Comparing empirical performance, we indeed found significant improvement over GPTQ. This was our process. Our apologies for not making this clear (and we will be sure to fix this in the revision!).
>
> We also emphasize that after optimizing our pipeline (as we show in the response to the Reviewer mcuu) we uniform improvements over SOTA for all rates (including 4 bits).
>
> Q3: $\frac{2\pi e}{12}$ is common in information-theoretic literature (2pi e comes from differential entropy of Gaussians and 1/12 from variance of a uniform quantization noise). But we agree that outside of IT this looks sloppy and annoying, so we will fix this too.
>
> Regarding performance of entropy coding, the concern about GPU kernel efficiency is completely valid. While upcasting to bf16 from a standard fp8/nvfp4/int4 storage is very simple entry-wise operation, decoding entropy-coded (compressed) weights requires a lot more processing .Note that  even a very simple (static table) rANS decompressor is sufficient for our purposes, which can be really fast. For example, already on A100 the https://github.com/facebookresearch/dietgpu reports 500+ GB/sec decoding speed. This clearly saturates the PCIe bandwidth (64 GB/s), thus showing usability of entropy-coding for accelerating model weight CPU offloading (weights stored uncompressed in HBM). If one wants storing compressed weights in HBM (and decodes them to SRAM) then the  HBM bandwidth (which is >1500+ GB/sec) cannot be saturated, but it’s still useful in modes when it is not a binding constraint (e.g. during prefill). Furthermore, on Blackwell+ there’s hardware support for compression.
>
>
> Nevertheless, we totally admit that overlapping decompression with other operations is a non-trivial kernel engineering task.
>
> Minor problems: we apologize for overlooking these works with mixed quantization rate. We will include them in the camera ready version. We agree with missing a factor in equation 3.1. We will also fix the citation format.

---

> > ### Author Rebuttal · Reviewer_x7Tw · 2026-04-03
> >
> > Thank you for the response!
> >
> > Most of my concerns are resolved by the authors. I agree that estimating the exact $R_{\mathrm{GPTQ}}$ and $R_{\mathrm{WaterSIC}}$ for a realistic distortion $D$ is an open problem, and making an optimized decompression GPU kernel for the proposed compression format is non-trivial. I think these do not limit the merits of this paper, and I would like to recommend acceptance. I have increased my rating (4->5).

---

### Official Review · Reviewer_P3YR · 2026-03-09

**Soundness:** 4
**Presentation:** 3
**Significance:** 4
**Originality:** 4
**Overall Recommendation:** 5
**Confidence:** 4

**Summary:**

This paper proposes WaterSIC, a post-training quantization (PTQ) method for linear layers that performs column-wise rate allocation inspired by information-theoretic waterfilling. The core thesis is that commonly used PTQ schemes (including GPTQ-style approaches) implicitly impose near-uniform bit allocation across columns, which can be far from optimal under a rate–distortion lens. Under an i.i.d. Gaussian weight model, the authors derive an information-theoretic benchmark and show that WaterSIC achieves performance within a small constant gap (reported as 0.255 bits) from the limit. Empirically, WaterSIC improves the perplexity–bitrate frontier on representative LLMs in a no-finetuning regime, indicating that rate allocation itself can yield nontrivial gains even with simple quantizers.

**Compliance With Llm Reviewing Policy:**

Affirmed.

**Key Questions For Authors:**

1. How sensitive is WaterSIC to the choice of distortion proxy (e.g., Hessian approximations, calibration data selection), and can the method degrade on certain layers (e.g., attention projections vs. MLP) if allocation is misestimated?
2. Can the authors provide a more standardized “effective rate” accounting across baselines (and across their own variants), ideally tied to an operational codec interpretation?

**Limitations:**

1. The near-optimality claims are derived under an i.i.d. Gaussian weight model, which may not capture real LLM weight statistics and layer-wise heterogeneity; thus, the theoretical gap bound may not reflect practical worst cases.
2. The paper does not quantify deployment-critical aspects such as kernel efficiency, end-to-end latency/throughput, and packing/format constraints under mixed per-column bit-widths, which could limit real-world benefits despite better perplexity–bitrate curves.

**Strengths And Weaknesses:**

Strengths
-  The paper identifies a concrete structural weakness of standard PTQ—uniform (or effectively uniform) bit allocation—then ties it to a rate–distortion argument with a clean algorithmic consequence (waterfilling-like per-column rates).
- Results support the claim that adaptive allocation improves the quality–rate trade-off, and the method appears competitive without finetuning, which is a meaningful operating point.
- The theoretical framing is not decorative, but it directly yields a practical procedure and a quantitative notion of near-optimality.
- The paper focuses on a specific, impactful component of LLM compression (linear-layer PTQ) and advances it with a coherent, well-argued method.

Weaknesses
- The strongest guarantees hinge on an i.i.d. Gaussian weight model; real transformer weights are structured, heavy-tailed, and layer-dependent. The empirical success is encouraging, but the theoretical near-optimality does not directly transfer.
- Mixing entropy-based rate notions with log-cardinality / nominal bit-width makes strict apples-to-apples comparisons harder, especially when claiming proximity to an information-theoretic limit.
-

---

> ### Author Rebuttal · Authors · 2026-03-31
>
> Q1. The point about calibration sensitivity is well-taken. While there exists prior work (arxiv:2405.20835) that asserts GPTQ is rather insensitive to the choice of calibration data, we unfortunately did not have time to reproduce those claims. To assess the stability of WaterSIC, though, we conducted the following experiments.
>
> We focused on the quantization of the Llama3.2-1B model into 2 bits.
> We vary the calibration set choice between the train of WikiText2 and the 1T sample of RedPajamas. To ensure fair comparison, we used the same number of sequences in both datasets(1189) equal to the size of the whole train of WikiText2 when split with the context length of 2048.
> Additionally, we vary the choice of the finetuning set between not doing finetuning at all, using WikiText2, RedPajamas and C4. Again, for all finetuning sets we use the 1189 sequence of length 2048.
> Finally, we evaluate both WikiText2 test PPL and C4 test PPL in the following tables:
>
> | Finetune   | W2 PPL | C4 PPL |
> |------------|--------|--------|
> | None       | 30.16  | 29.88  |
> | WikiText2  | 14.91  | 23.65  |
> | RedPajamas | 18.42  | 22.28  |
> | C4         | 18.43  | 20.37  |
>
> **Table 1: Calibration on RedPajamas**
>
> | Finetune   | W2 PPL | C4 PPL |
> |------------|--------|--------|
> | None       | 16.19  | 39.39  |
> | WikiText2  | 13.46  | 28.86  |
> | RedPajamas | 15.11  | 24.37  |
> | C4         | 15.10  | 21.96  |
>
> **Table 2: Calibration on WikiText2**
>
> We can see that finetuning on any combination of datasets improves both WikiText2 and C4 PPL.
>
> Q2. Reviewer is correct in questioning whether one should compare log-cardinality/nominal bit-width with entropy-coded rate. The answer is that one generally should not. Exception is with vector quantizers, as those both have similar computational overhead and strive to solve the same problem as entropy coding (group entries in the weight matrix and compress them jointly). Thus, for this reason, we believe that comparing *scalar* but entropy-coded quantizers (WaterSIC and Huffman-GPTQ) to “nominal bitwidth of QuIP#,PV-tuning, NestQuant, QTIP etc” is fair, which is the reason we included them in the paper. We should have emphasized better the secondary point (in addition to variable rate) that we were trying to make: scalar + entropy-coding basically rips all the benefits of vector quantization, while building on reusable hardware blocks, and also allowing (as WaterSIC demonstrates) almost continuous variation of the rate allocation.
>
> Regarding Gaussianity, excellent question.
>
> It is true that our theoretical studies only treat the special case of iid Gaussian weights. We think, however, that this special case is rather representative for several reasons. First, one may use randomized rotations to “Gaussianize” various outliers in weights. Second, the outliers, while quite common in early LLMs, appear to vanish with modern optimizers, improved layernorms and learning rate schedules [1]. We computed a Kolmogorov-Smirnov distance (max discrepancy between empirical and conjectured CDFs) to MLE Gaussian and Laplace (as suggested) and found out the following for Llama 2-7B:
>
>
> | Weight type | mean D_gauss | mean D_laplace | Gauss better (layers) |
> |---|---:|---:|---:|
> | w1 | 0.63% | 4.13% | 32 / 32 |
> | w2 | 0.32% | 4.24% | 32 / 32 |
> | w3 | 0.33% | 4.26% | 32 / 32 |
> | wq | 2.96% | 2.99% | 23 / 32 |
> | wk | 3.16% | 2.66% | 16 / 32 |
> | wv | 0.84% | 3.93% | 30 / 32 |
> | wo | 0.93% | 3.99% | 30 / 32 |
>
>
> So w1, w2, w3, wv, wo are “very Gaussian” (max deviation < 1%), while wk, wq are a little worse and sometimes Laplace fits their distribution better. Similar results for other Llamas: the heaviest layers w1-w3 are Gaussian, wk and wq are often more Laplace. We have attached plots of weight histograms and corresponding best-fitting Gauss and Laplace distributions on some layers of Llama3.2-1B in “gaussianity.pdf” at [2]. [2] also contains updated ppl/bitrate plots
>
>
> [1] Jungwoo Park, Taewhoo Lee, Chanwoong Yoon, Hyeon Hwang, and Jaewoo Kang. 2025. Outlier-Safe Pre-Training for Robust 4-Bit Quantization of Large Language Models. arXiv:2506.19697. https://arxiv.org/abs/2506.19697
>
> [2] https://www.dropbox.com/scl/fo/ig1d3dbrsyigrc0d2tkh4/ABzwMGIT6j1Veomr8ktHgUA?rlkey=u082bjhi7z32h45roz07ljto8&st=cs3rhhny&dl=0
>
> Regarding the speed/latency – that’s a fair and important limitation, which we admitted in the paper.
> We have a discussion on this topic in the response to Question 1 of Reviewer QwRE.

---

> > ### Author Rebuttal · Reviewer_P3YR · 2026-04-04
> >
> > My cocerns are fully resoloved. I think that this manscript is sufficient to be presented.

---

### Official Review · Reviewer_mcuu · 2026-03-11

**Soundness:** 3
**Presentation:** 3
**Significance:** 3
**Originality:** 3
**Overall Recommendation:** 4
**Confidence:** 4

**Summary:**

The paper brings information-theoretic insights and analysis to the problem of post-training quantization. They propose a waterfilling/successive interference cancellation (SIC) based bit allocation method for quantizing linear layers. Waterfilling is a classical power allocation technique used for communication over parallel independent channels. The authors interpret the columns of a linear layer as parallel channels and use waterfilling to allocate bits across them. Since these channels are not independent, they incorporate a successive interference cancellation strategy. The paper first analyzes a single-layer neural network, showing that the proposed method outperforms standard entropy-coded GPTQ and achieves performance within a finite gap of the theoretical limit under the considered distortion metric. The paper then applies the proposed approach to large-scale language models and demonstrate empirical improvements over several existing post-training quantization baselines.

**Compliance With Llm Reviewing Policy:**

Affirmed.

**Final Justification:**

I keep my positive score.

**Key Questions For Authors:**

* Discussion on the extension of the theoretical parts to more general/broader settings would be interesting.
* Comparisons with optimization-based post-training quantization methods would help better access the competitiveness of the proposed approach.
* The method relies heavily on activation covariance estimated from calibration data to determine the bit allocation. It is unclear how sensitive the approach is to distribution shift between calibration data and inference-time inputs. Baseline methods may be similarly sensitive to the distribution shift. Their relevant performance is not quite clear.

**Limitations:**

Yes

**Strengths And Weaknesses:**

Strengths:

* The paper brings an information-theoretic perspective to post-training quantization, which is conceptually interesting. The use of waterfilling and successive interference cancellation, while classical in information theory and communications, is a creative way to interpret and analyze bit allocation for neural network quantization.
* The paper includes both theoretical analysis and large-scale empirical experiments on modern language models, and the reported results appear promising.

Weaknesses:
* There is a noticeable gap between the theory and the empirical setting. The theoretical analysis is developed for a single linear layer with an MSE objective, whereas real neural networks consist of multiple interacting layers with nonlinear activations, and evaluation is based on metrics such as perplexity. It is unclear how strongly the theoretical guarantees translate to the full-model setting, despite the empirical results looking promising.
* The empirical evaluation does not include comparisons with optimization-based post-training quantization methods. While such methods may have higher computational cost, they represent strong baselines and would help better assess the competitiveness of the proposed approach.
* The method relies heavily on activation covariance estimated from calibration data to determine the bit allocation. It is unclear how sensitive the approach is to distribution shift between calibration data and inference-time inputs.

---

> ### Author Rebuttal · Authors · 2026-03-31
>
> We agree that end-to-end theory would be more interesting, but we (given that most PTQ algorithms right now are largely solving a layer-wise task), we believe that info-theoretic analysis of that setting was a reasonable start. However, in the revision, we attempted to optimize layerwise loss metric with some inter-layer optimization.
>
> Since the original submission, where we used standard GPTQ hessian, we have improved our choice of the loss metric by adding two ideas: 1) attention weighting (for W_K, W_Q, W_V layers we compute covariance matrix of activations but weight tokens proportional to the attention they receive from others); 2) doing golden ratio search between using GPTQ hessian and Qronos/QA-LDLQ drift-corrected one. Overall, adding these allowed us to reduce, for example, PPL on Llama-3.2-1B for 3 bits from 10.71 (in the submission) to 10.57:
>
> | Model | Bitwidth | Original | Improved | Δ |
> |-------|----------|----------|----------|-------|
> | Llama-3.2-1B | 2.0 | 18.40 | 16.19 | −2.21 |
> | Llama-3.2-1B | 3.0 | 10.71 | 10.57 | −0.14 |
> | Llama-3.2-1B | 4.0 | 9.94 | 9.92 | −0.02 |
> | Qwen3-8B | 2.125 | 11.65 | 11.37 | −0.28 |
> | Qwen3-8B | 3.125 | 10.10 | 10.03 | −0.07 |
> | Qwen3-8B | 4.125 | 9.83 | 9.79 | −0.04 |
>
> We also have some initial results for the Llama-3-70B quantization on 4 bits. We ran our algorithm on only 2 H100s, without post-quantization finetuning. We evaluated WikiText2 PPL for the context length of 2048:
>
> | Method | WikiText2 PPL |
> |--------|---------------|
> | BF16 | 2.86 |
> | RTN | 3.6 |
> | QuIP | 3.4 |
> | GPTQ | 3.3 |
> | AWQ | 3.3 |
> | OstQuant | 3.19 |
> | NestQuant | 3.14 |
> | **WaterSIC (ours)** | **3.06** |
>
> In our initial submission we emphasized performance of the layerwise algorithm, because it was our goal to obtain the most competitive linear layer quantizer (which is also useful in many other areas of ML, e.g. fast high-dimensional regression in online ads and HFT finance, vector DBs etc). However, given the requests of reviewers we performed a quick test of the end-to-end finetuning using the following strategy.
>
> After quantization, we finetune the continuous parameters (row rescalers $T$ and column rescalers $\Gamma$) via KL-divergence distillation from the unquantized teacher model. The integer codes Z and all other model parameters are frozen; only $T$ and $\Gamma$ are trainable (~0.25% of total parameters).
>
> We use AdamW (no weight decay) with cosine annealing from $lr = 5 \cdot 10^{-3}$ to $5 \cdot 10^{-6}$ over 5 epochs on the whole finetuning set. We experiment over the choice of the finetuning set, setting WikiText2 train split as the standard choice. To preserve deployment fidelity, $T$ and $\Gamma$ are rounded to bf16 precision via straight-through estimation during training.
>
> Even in this simple, unoptimized setting, the results are quite impressive, especially on lower bitrates:
>
> | Model | Type   | 1-bit   | 2-bit   | 3-bit   |
> |-------|--------|---------|---------|---------|
> | Llama 3.2 1B | Non-FT | 82.44   | 16.19   | 10.57   |
> |               | FT     | 30.4718 | 13.5596 | 10.4461 |
> | Llama 2 7B    | Non-FT | 22.43   | 6.5728  | 5.6468  |
> |               | FT     | 12.0319 | 6.2873  | 5.6301  |
> | Qwen 3 8B     | Non-FT | 25.5925 | 11.3706 | 10.0317 |
> |               | FT     | 14.6648 | 10.1833 | 9.7307  |
> | Llama 3 8B    | Non-FT | 47.1541 | 8.9319  | 6.6520  |
> |               | FT     | 18.9734 | 8.0556  | 6.5971  |
>
> In addition, we compare our results with other methods that also include finetuning for 2-bit Llama-2-7B. Note that these results are for a context length of 4096, while we used a model quantized with calibration at a context length of 2048 and then finetuned it at a context length of 4096. With more time, we would be able to improve these results by calibrating at the same context length as well.
>
> | Method | Bitwidth | WikiText-2 PPL |
> |---------------|----------|----------------|
> | AQLM | 2.02 | 6.64 |
> | QuIP# | 2.01 | 6.19 |
> | DB-LLM | 2.01 | 7.23 |
> | PV-Tuning | 2.02 | 5.84 |
> | WaterSIC (ours)| 2.00 | 5.87 |
>
> Note that the perplexity differs from results in the previous table because this evaluation is on a different context length (4096 vs 2048).
>
> For the camera ready we will include more detailed simulations, especially dependence on the finetuning dataset. We attach the ppl/bitrate plots at [1]
>
> We discuss the calibration sensitivity in great detail in response to Question 1 of Reviewer P3YR.
>
> [1] https://www.dropbox.com/scl/fo/ig1d3dbrsyigrc0d2tkh4/ABzwMGIT6j1Veomr8ktHgUA?rlkey=u082bjhi7z32h45roz07ljto8&st=cs3rhhny&dl=0

---

> > ### Author Rebuttal · Reviewer_mcuu · 2026-04-06
> >
> > I appreciate the authors for providing additional experimental results.

---

### Official Review · Reviewer_QwRE · 2026-03-13

**Soundness:** 3
**Presentation:** 3
**Significance:** 3
**Originality:** 3
**Overall Recommendation:** 5
**Confidence:** 1

**Summary:**

This paper addresses the problem of compressing a full-precision linear layer into a low-precision version. The authors first assume that each row of weight matrix is iid gaussian random vector with zero mean. Under this condition, the authors show that GPTQ algorithm has a large gap to information theoretical (IT) limit. To tackle this problem, WaterSIC algorithm is proposed, and it is shown to be within rate gap of 0.255 bit to IT limit.

**Compliance With Llm Reviewing Policy:**

Affirmed.

**Final Justification:**

The authors have fully addressed my concerns. I have increased my score (4->5).

**Key Questions For Authors:**

Q1) Does using different quantization rate for different weight columns not disturb implementing efficient GPU kernel?

Q2) What does the superscript '*' in line 182 mean? It should be described explicitly.

**Limitations:**

Yes

**Strengths And Weaknesses:**

1) Soundness : It is unclear that usage of different quantization rate for each column can be efficiently implemented in current GPUs. Using the proposed method, the memory access pattern may become complicated, because simple path of reading weights with a uniform stride and unpacking them in a fixed way breaks down. As a result, there may be additional computational bottleneck.

2) Presentation : This paper is mathematically heavy, but it is structured.

3) Significance : This paper address an important problem.

4) Originality: This work first propose to use different quantization rate for different weight columns.

---

> ### Author Rebuttal · Authors · 2026-03-31
>
> We thank the reviewer for taking time to read our paper and provide feedback! Here are our responses to the questions:
>
> Q1. The concern about GPU kernel efficiency is completely valid. While upcasting to bf16 from a standard fp8/nvfp4/int4 storage is very simple entry-wise operation, decoding entropy-coded (compressed) weights requires a lot more processing .Note that even a very simple (static table) rANS decompressor is sufficient for our purposes, which can be really fast. For example, [1] achieves 500+ GB/sec decoding speed on A100 GPU. This clearly saturates the PCIe bandwidth (64 GB/s), thus showing usability of entropy-coding for accelerating model weight CPU offloading (weights stored uncompressed in HBM). If one wants storing compressed weights in HBM (and decodes them to SRAM) then the  HBM bandwidth (which is >1500+ GB/sec) cannot be saturated, but it’s still useful in modes when it is not a binding constraint (e.g. during prefill). Furthermore, on Blackwell+ there’s hardware support for compression. Nevertheless, we totally admit that overlapping decompression with other operations is a non-trivial kernel engineering task.
>
> Q2. The notation {0,1}^* simply denotes “variable length binary string” (could be of length zero). One can think of it as a tuple (n, arr), where array arr consists of n bits
>
> [1] https://github.com/facebookresearch/dietgpu
> Nevertheless, we totally admit that overlapping decompression with other operations is a non-trivial kernel engineering task.

---

> > ### Author Rebuttal · Reviewer_QwRE · 2026-04-01
> >
> > The authors resolved the questions clearly.

---

### Decision · Program_Chairs · 2026-04-30

**Decision:**

Accept (spotlight)

**Comment:**

There is a consensus that this paper gives a clear and original contribution by using a rate-distortion approach to post-training quantization.  The theory was praised as being substantive and meaningful, and the empirical performance was assessed as being strong, improving the trade-off of quality/rate/perplexity over baselines.  The rebuttal has generally resolved the technical concerns raised, such as those around calibration sensitivity and implementation caveats.